



# Pre-launch calibration and validation of the Airborne Hyper-Angular Rainbow Polarimeter (AirHARP) instrument

Brent A. McBride[2,3], J. Vanderlei Martins[1,2], J. Dominik Cieslak[1,2], Roberto Fernandez-Borda[1,2], Anin Puthukuddy[1,2], Xiaoguang Xu[1,2], Noah Sienkiewicz[1,2], Brian Cairns[4], and Henrique M. J. Barbosa[1]

[1]Department of Physics, University of Maryland, Baltimore County, MD, USA
[2]Earth and Space Institute, University of Maryland, Baltimore County, Baltimore, MD, USA
[3]Science Systems and Applications, Inc., Lanham, MD, USA
[4]NASA Goddard Institute for Space Studies, New York, NY, USA

*Correspondence to*: Brent A. McBride (mcbride1@umbc.edu)

**Abstract.** The Airborne Hyper-Angular Rainbow Polarimeter (AirHARP) is a new imaging polarimeter instrument, capable of sampling a single Earth target from up to 120 viewing angles, in four spectral channels, and three linear polarization states across a 114° field of view. AirHARP is telecentric in the image space and simultaneously images three linear polarization states with no moving parts. These two aspects of the design allow for a simple and efficient quantitative calibration. Using coefficients derived at the center of the lens and the detector flatfields, we can calibrate the entire AirHARP sensor in a variety of lab, field, and space environments. We describe the calibration process for the HARP family of polarimeters using AirHARP pre-launch data. We show that this telecentric calibration technique yields a 0.25% RMS degree of linear polarization (DOLP) accuracy in all channels for pixels around the detector center. To validate across the FOV, we compare our multi-angle reflectance and polarization data with the Research Scanning Polarimeter (RSP) over targets sampled during the NASA Aerosol Characterization from Polarimeter and Lidar (ACEPOL) campaign. For the majority of angles and targets used in the intercomparison, RSP and AirHARP agree better than 1% in reflectance and DOLP. Therefore, our calibration successfully transfers nadir coefficients to different FOVs, given ambient challenges. This calibration technique makes the HARP design attractive for new spaceborne climate missions: HARP CubeSat (2020-2022), HARP2 (2024-) on the NASA Plankton-Aerosol-Cloud-ocean Ecosystem (PACE), Atmosphere Observing System (AOS) and beyond.



# 1 Introduction

Aerosols and their effect on clouds are one of our largest climate challenges. These particles are difficult to model and measure from satellite. Some aerosols are irregularly shaped, absorbing, and dimly reflecting, and others are spherical and efficient at scattering sunlight. They are transported across the globe from a variety of source regions, perturb the boundary layer, and interact with clouds in different ways depending on their location in the atmosphere and composition. Aerosols can control how long clouds last, how bright they are, and when they will precipitate (Boucher et al. 2013). This complexity makes it difficult to estimate the impacts of clouds and aerosols on our climate system. However, this uncertainty drives innovation and instrument development. Satellite measurements with a range of spectral, angular, spatial, and polarized capabilities can improve how we measure these properties at global scales (NASA 2018). Instruments that combine these features, called multi-angle polarimeters (MAPs), may make considerable enhancements to our climate record in this direction (Dubovik et al. 2019). They are highly compatible with current instruments, expand the information content possible in a single measurement, and can be designed to small and cost-effective form factors. Recent studies show that microphysical retrievals done on multi-angle polarimetric (MAP) data are highly attractive for future missions and improving our knowledge of microphysical properties (Mishchenko et al. 2004, Knobelspiesse et al. 2012, Stamnes et al. 2018, Remer et al. 2019). Of these, those that sample with less than or equal to 3% in absolute radiometric calibration and 0.5% *degree of linear polarization* (DOLP) uncertainty are optimal (NASA 2015, NASA 2021).

Over the past decade, several research teams have demonstrated a variety of effective MAP designs in aircraft campaigns and lab calibrations. Prominent MAP instruments include the Research Scanning Polarimeter (RSP, Cairns et al. 1999), the SPEX (Hasekamp et al. 2019), Airborne Multi-angle Spectro-Polarimetric Imager (AirMSPI, Diner et al. 2013), and the Hyper-Angular Rainbow Polarimeter (HARP, Martins et al. 2018). The RSP is a 14mrad single-pixel scanner that measures polarized radiation from a target pixel at up to 150+ viewing angles. The RSP measures these angles across nine spectral channels from 410-2200nm. Its narrow viewing angle density (0.8°), together with high polarimetric uncertainty (~0.002 in DOLP), allows for a near-seamless reconstruction of the scattering profile of any ground target. RSP measurements paved the way for new cloud and ice property retrievals (Sinclair et al.



2021, van Diedehoven et al. 2013), ocean color (Chowdhary et al. 2012), and improved cross-comparisons with other instruments (Knobelspiesse et al. 2019, van Harten et al. 2018, Smit et al. 2019). The RSP represents one possible MAP design but there are others that take advantage of spectral information and modulation of the polarized signal. The SPEX instrument is a hyperspectral multi-angle polarimeter capable of measuring a ground target from five to nine viewing angles over 109 spectral channels (400-

800nm). SPEX selects wavelengths using an internal diffraction grating and uses spectral modulation to de-convolve total radiance and DOLP signals from the measurements. SPEX measurements may narrow uncertainty in aerosol microphysical retrievals of single scattering albedo, size, shape, and refractive indices, beyond the capabilities of current space platforms (Hasekamp and Landgraf 2007, Hasekamp 2019). Also, highly accurate (~0.002 in DOLP), SPEX was one of the polarimeters contributed to the

NASA PACE mission specifically for new aerosol science (Werdell et al. 2019). The MSPI instrument measures the incident polarization state of a target using photoelastic modulation (Diner et al. 2013). MSPI tracks the same point on the ground with a programmable gimble that locks into specific view angles (*step-and-stare*). In a separate sampling mode, MSPI scans its two-dimensional FPA over a wide range of scattering angles (*continuous sweep*). The MSPI concept was optimized into the Multi-Angle

Imager for Aerosols (MAIA), a space mission that will characterize air pollution over city targets (Diner et al. 2018). The MSPI team reports <0.005 DOLP uncertainty for all spectral channels, which is achieved and further improved by aggregating pixels (van Harten and Diner, 2015).

    This paper discusses the fourth instrument mentioned above, the HARP, a wide field, Earth-observing modern MAP that is capable of highly-resolved, highly-accurate climate measurements. This

work focuses on AirHARP, the aircraft version of the HARP design. This work will discuss the optics of the instrument, how it combines the strengths of the above instruments, and how our calibration process maintains high measurement accuracy in lab and in the field. In this paper, we introduce the AirHARP instrument (Section 2), step through the full quantitative calibration process in detail (Section 3), and discuss validation studies done in the lab and on flight data (Section 4). We close in Section 5 with a

discussion of limitations and look ahead to the HARP CubeSat satellite payload and the HARP2 deployment on-board the NASA PACE mission in 2023.



## 2 The Airborne Hyper-Angular Rainbow Polarimeter (AirHARP)

The AirHARP instrument is a wide field-of-view imaging polarimeter, shown in Figure 1. AirHARP samples Earth targets at four optical channels: 440 (14), 550 (12), 670 (18), and 870 (37) nm. These four channels are selected passively using a custom stripe filter on top of a charge-coupled device (CCD) detector FPA. The stripe filter distributes the four AirHARP channels across 120 distinct regions of the detector, called *view sectors*. The 670nm band covers 60 of these view sectors and the other 60 are split equally across the other three bands. Each detector is covered by a linear polarizer, set at a unique angle. These detectors are focused at the three output ports of a custom beamsplitter. This design decouples the incident polarization into orthogonal S-and P-states at each FPA. In post-processing, co-located information in each detector is combined to reproduce the Stokes parameters (I, Q, U) of the original beam. Simultaneous polarization imaging by the three detectors, across four spectral channels, allows high polarization accuracy with no moving parts.

The core of AirHARP's polarization sensitivity is the custom Phillips prism, shown in Figure 2. While this prism is typically designed to split colors (Alternative Vision), the AirHARP prism splits the polarization content of the original signal into the three AirHARP detectors. The prism is made of three individual glass elements, A, B, and C, of equal index of refraction. The prism is a major component of an optical train that contains eight other sequential elements and a 114° wide field front lens. These lenses are optimized for optical throughput into the prism and create an imaging sensor that is telecentric in the image space. This feature is critical to our calibration and will be discussed in later sections. Most importantly, this refractive design allows wide-field of view measurements in a 3U CubeSat housing (10x10x30cm).

The modified Phillips prism alters each detector's light path in a specific way. The incident beam first enters the prism at the front face of Element A and meets the boundary between Elements A and C. A custom splitting coating at the boundary reflects 33% of this light back into Element A. Reflections like this reduce P-polarization and preserve S-polarization. Transmissions do the reverse. To boost the efficiency of the final polarization measurements, we align the Detector A polarizer with this S-polarization state, which is defined as 0°. The light path defined by the wide FOV front lens, optical train,



the prism, the 0° polarizer, and the Detector A FPA is called *Sensor A*. The convention of our polarimetric

calibration is relative to this Sensor. The two other light paths through the optics define Sensors B and C.

The light that passes through this boundary contains primarily P-polarized light. At the interface

between Elements B and C, another thin-film coating splits the light intensity 50%-in-reflection and 50%-

in-transmission. So far, the polarization content of this beam has changed by a transmission through the

Element A-C interface and a reflection at the B-C interface. Therefore, the light incident on Detector C is

a weak mixture of S- and P-states. The detector polarizer can be set at any angle with minimal effect on

polarization efficiency. During optimization testing, we found the best orientation to be 90° for the

Detector C polarizer, and likewise 45° for the Detector B polarizer. This 45° relative separation between

the polarizers is optimal to discriminate measured states of polarization in our design (Tyo et al. 2006).

Sensors B and C each account for 33% of the intensity of the incident beam, as well. Therefore, the

AirHARP optics splits the incident light intensity equally among the three Sensors, each Sensor images a

spatially identical scene, and each Sensor is sensitive to a different angle of polarized light.

Light that passes through the prism and detector polarizer is categorized by a custom

interferometric filter on the detector surface. Each detector pixel maps to a specific spectral band, defined

by one of the 120 view sectors. AirHARP produces a pushbroom of a ground scene in a single view sector

by flying over the scene and acquiring images one after the other. The co-located information from

multiple view sectors can provide high angular coverage on the cloudbow at 670nm (McBride et al. 2020)

and multi-angle sampling of aerosol optical, shape, size, and loading properties (Hasekamp and Landgraf

2007, Wu et al. 2015, Puthukuddy et al. 2020, Meng et al. 2021) and atmospheric correction (Frouin et

al. 2019). Aerosol and cloud properties retrieved by AirHARP (and future HARP instrument)

measurements may complement our existing climate record and advance our understanding of climate

change uncertainties, feedbacks, and forcings (Boucher et al. 2013).

The AirHARP instrument and spaceborne version, the HARP CubeSat, were funded by the NASA

Engineering Science and Technology Office InVEST program as a demonstration of advanced,

miniaturized Earth science technology for future satellite missions. The HARP CubeSat recently

completed a two-year mission in the 425km apogee orbit of the International Space Station. AirHARP

was built specifically for science aircraft, like the NASA B-200 and ER-2, and demonstrated the HARP



design and capabilities in field campaigns before and during the CubeSat mission. AirHARP flew successfully in two NASA aircraft studies in 2017: the Lake Michigan Ozone Study (LMOS) in the summer (McBride et al. 2020) and Aerosol Characterization from Polarimeter and Lidar (ACEPOL) in

the fall (Knobelspiesse et al. 2020). In the early 2020s, a third, highly advanced version of the HARP concept, HARP2, will fly on the NASA PACE spacecraft for a mission lifetime of three years (McBride et al. 2019).

While the calibration discussed in later sections is the general scheme for any of the HARP instruments, plots, tables and figures correspond the AirHARP instrument, unless otherwise noted.

Whenever the term HARP is used without an "Air" prefix or "CubeSat" suffix, it is in reference to a general HARP design.

## 3 Calibration Scheme for HARP Instruments

### 3.1 Detector Specifications and Background Correction

The calibration process of the AirHARP begins at the detector level. The AirHARP detectors are

monochrome CCDs with a 4 megapixel active FPA (Semiconductor Components Industries 2015). Relevant properties, such as quantum efficiency, read noise, and dark current, are given in Table 1. The typical image taken by the AirHARP detectors are shown in Figure 3a. The detector stripe filter creates the cross-track striping in the images seen below. The far left and right detector pixels are masked, which defines the active science area of the FPA. The pixel values in these areas are compatible with a *dark*

*image*, which is a snapshot taken when the entire FOV is blocked from illumination, shown in Figure 3b.

The first step in the AirHARP calibration begins at the detector level. Detectors generate a stable electrical bias across the FPA when they operate, which must be removed before science analysis. To account for this, we block all illumination from reaching the front lens (i.e. with a lens cap or internal shutter) and take 10 or more sequential images in each detector. These images are averaged together into

a *dark template*. Creating this template image is called the *standard process* in this work going forward. The typical distribution of the dark template is given in Figure 3b. The region of lower pixels on the left-hand side of the image is typical of CCDs and occurs as photoelectrons move toward the serial register.





A typical dark signal for the AirHARP detectors is 40 counts when operating at room temperature. In general, the background correction is as follows:


$$DN_{BC} = DN_{raw} - DN_{dark},$$ (1)

where $DN_{BC}$ is the background corrected image *digital numbers* or counts, $DN_{raw}$ are the raw image counts, and $DN_{dark}$ represents the dark template counts. Whenever the term *raw* is used it refers to any

HARP image, whereas subscripts other than *raw* describe an image captured in a different environment. Furthermore, counts may be called *analog-digital units* (ADU) in this work, if relevant. In sensitivity studies on AirHARP dark image data, the dark counts do not depend on integration time, but are sensitive to operating temperature.

Both AirHARP and HARP CubeSat have an internal shutter, which is actuated for on-orbit dark

captures. This shutter does not contribute to polarization imaging and defaults to an open configuration outside of the optical path as a fail-safe. If we cannot take dark captures on-orbit or during field campaigns for any reason, we can create a *synthetic dark* by scaling a normalized dark template from the lab by the measured counts in the vignetted areas of a live data capture (cross-track pixel indices 0-200 or 1848-2048):


$$DN_{dark} = \delta \cdot DN_{raw\,[0-200,1848-2048]}$$ (2)





where $DN_{dark}$ is the estimated dark image counts and $\delta$ is the normalized dark template image. $DN_{raw\,[0-200,1848-2048]}$ represents pixels in the vignetted area of a raw image capture (similar to Figure 3a). Eq. (2) creates a full-field dark image for each sensor that is used in the following calibration steps

and in the Level 1B processing of AirHARP flight imagery. If Eq. (2) is required, i.e. the $DN_{dark}$ here is substituted into Eq. (1). This technique is currently used to correct AirHARP L1B datasets in Version 002 and accounts for the possibility of internal shutter failure on-orbit.

In the following sections, we limit our discussion to the 670nm channel, unless otherwise noted. Similar performance for the other three channels can be found in official ancillary basis documents
(ACEPOL Science Team, 2017).

### 3.2 Flatfielding

Next, we characterize the pixel-to-pixel relative response of each detector. Any system with sequential optical elements will vignette photons toward the edge the FPA. Individual pixels may have a relative differential gain, as well. Both effects must be corrected. To account for this, images are taken of
a homogenous target in a process is called *flatfielding*. Integrating spheres are typical sources. They create uniform illumination over their aperture and can depolarize the output to a level below 0.5% in visible wavelengths (McClain et al. 1994). Therefore, any heterogeneity in the images is due to the instrument, not the source. The "Grande" 101.6cm integrating sphere at NASA GSFC is a suitable source, with a port fraction <5% and sphere multiplier >10 (PerkinElmer). We also use a portable LED hemisphere at UMBC
with similar attributes, during field campaigns or between GSFC calibrations. To form the flatfield template, the full-FOV of the AirHARP instrument images the aperture of an illuminated integrating sphere, at an integration time where all channels are below saturation. The images are full-size, full-resolution and resemble Figure 3a. A template image, created using the standard process, is background corrected via Eq. (1) and (2). This template is then interpolated by a smoothing algorithm, row-by-row.
This step captures the structure of vignetting and other potential artifacts, such as optical etaloning and defects on the detector surface (andor.oxinist.com).

Figure 4b shows a cross-track line-cut for several 670nm view sectors: +27º (red), +14º (blue), nadir (grey), -8º (green), and -20º (magenta) are shown. The x-axis is cross-track pixel index, starting



with 200 at the image far left and 1800 on image far right. This region represents the active science area

of the detector. The y-axis is detector counts (ADU). Each curve is artificially offset by +/- 500 or 1000 ADU for clarity, though the nadir curve corresponds directly to the y-axis values. Without these offsets, each of the curves would overlap with the nadir curve. The counts data for each row is smoothed using a 15-pixel sliding window average (black). The smoothing process also captures other stable artifacts in the images (i.e. oscillations due to optical etaloning), that can be removed as part of this correction. We repeat

this smoothing process for each channel and detector row until we arrive at a smoothed full-field template image, at the same size and resolution as the original data. We then normalize the smoothed signal of each channel by relevant pixels along the optical axis. This normalized, smoothed signal becomes the flatfield correction, *f*, for this channel and detector. Normalization is done so that the flatfield is scalable to any reflectance level in a field measurement. Each pixel in the FOV has a different value of *f*. The

optical axis is chosen specifically as the location of *f*=1 in order to simplify the later steps in the calibration process that also use optical axis pixels. We then apply the flatfield correction at the pixel-level:

$$DN_{flat} = \frac{DN_{BC}}{f}, \tag{3}$$

where f is the valye of the flatfield correction for that pixel, and the numerator of Eq. (3) is the same as Eq. (1). To verify the flat correction, we apply the flatfield to its generating dataset via Eq. (3). Figure 4c shows a histogram of the residuals after flatfielding all pixels in the same subset of view sectors as Figure 4b. The data point colors in Figure 4c map to the same view sector colors in Fig. 4b. The original signal is corrected down to signal-to-noise (SNR) variations at the 0.005 level for each view sector. Figure 4c

shows that this method is robust across the FOV and accurately removes all systematic artifacts in the data. Moreover, this correction creates a detector-specific flatfield *f* for each of the four AirHARP channels.

The flatfield serves another critical role in the AirHARP calibration. AirHARP optics are telecentric in the image space, and so all incident rays on the detector arrive at 0° angle-of-incidence

(AOI). This design prevents AOI-related artifacts in the images or dependency in the calibration



coefficients. Our flatfield represents the entire internal optical behavior of the system and simplifies our next calibration steps in the process. We can derive channel-dependent coefficients at any location in the FPA and spread that result to the rest of the FOV using the detector flatfields. This *telecentric technique* is the method used in the following steps of our calibration process. We also verify these coefficients

using lab techniques and across the full FOV using field data in Sections 3 and 4.

## 3.3 Non-Linear Correction

AirHARP sensors are commercial CCDs. They are subject to non-linearity in their analog-to-digital conversion (ADC). For very bright targets, like sunglint, the Earth's limb, or direct solar exposure, pixels may saturate at the top of the detector well (44,000 electrons or $2^{14}$ counts). Saturated pixels cannot

convert any extra photoelectrons to counts, but CCDs are known to have a non-linear gain near saturation and potentially at very low light levels (Semiconductor Components Industries 2015).

The detectors must have a well-characterized gain for accurate science retrievals, too. We characterize non-linearity by taking images of a stable source at a single illumination level. Each image is taken at a longer integration time than the last, and the testing ends when all sensors and channels are

saturated. To perform this test, the AirHARP instrument was placed ~1m from the entrance aperture of the NASA GSFC "Grande" sphere. The AirHARP detector integration times are set near 4ms to start. The integration times of each sensor are increased and images are taken until all three sensors and channels saturate. The stability of the source is tracked over the testing window using a current monitor. The standard process is used to form a template image at each integration time and for each detector. We take

a small pixel bin (~4x4) along the optical axis in the templates and plot those values against their integration times. This process is performed for each channel and sensor. An example for the 670nm channel is shown in Figure 5a, for the three AirHARP detectors (Sensor A in red, Sensor B in blue, Sensor C in green). Instead of using integration times, we plot detector counts against *integration lines*, which are the exposure settings used by the AirHARP timing board. Integration lines convert to times via 0.237

ms * (2117 – Lines). Larger integration line values correspond to shorter integration times. There is a monotonic, positive relationship between integration lines and detector counts, up until the saturation point, $2^{14}$ ADU. In this example, Sensor B saturates earlier than Sensors A or C, due to the ratio of



integration times between detectors. We then fit all counts in the linear region of the detector, defined by values < 3000 ADU. There is no evidence of low light non-linearity in these detectors. We then compare

this fit curve to the rest of the data:

$$DN_{corr} - DN_{flat} = n_0 DN_{flat}^2 + n_1 DN_{flat} + n_2,$$    (4)

where $DN_{corr}$ is the non-linear corrected counts data, $DN_{flat}$ is the counts data derived from Eq. (3), and

fit parameters $n_0$, $n_1$, and $n_2$ are free parameters. Eq. (4) is explicitly separated into two terms for trending of the non-linear coefficients. In our Figure 5b example, the residual ($DN_{corr} - DN_{flat}$) is the y-axis and the x-axis is $DN_{flat}$. The maximum non-linear deviation at 670 nm is 4% in Sensor A, found by taking the ratio $DN_{corr}/DN_{flat}$. This ratio agrees with the 6% non-linearity limit in KAI-04070 detector spec, and similar agreement is found for other channels and sensors. The three above tests occur before

any other step in the Level 1B processing pipeline for HARP data. We perform non-linearity early in the calibration pipeline to check detector-level anomalies, though other instruments, like MODIS, can be characterized for non-linearity during radiometric calibration (Aldoretta et al. 2019).

### 3.4 Relative polarimetric calibration

### 3.4.1 Theoretical description

After the images are corrected for background, flatfield, and non-linearity, and the detectors are mechanically co-aligned in the image space, the instrument is ready for quantitative polarization calibration. Relative states of polarization are converted to co-located counts in the three detectors and vice versa. The theory of our calibration is given in Fernandez-Borda et al. (2009), though a brief treatment of the scheme is discussed here.

The polarization state of a light beam is described by the Stokes column vector, which is a time-average (designated by the enclosing brackets) of the real and imaginary components of the electric fields (Jackson 1962):



$$S = \begin{bmatrix} I \\ Q \\ U \\ V \end{bmatrix} = \begin{bmatrix} \langle E_\parallel E_\parallel^* + E_\perp E_\perp^* \rangle \\ \langle E_\parallel E_\parallel^* - E_\perp E_\perp^* \rangle \\ \langle E_\parallel E_\perp^* + E_\perp E_\parallel^* \rangle \\ i\langle E_\parallel E_\perp^* - E_\perp E_\parallel^* \rangle \end{bmatrix}, \tag{5}$$


where $E_\parallel$ and $E_\perp$ the parallel (S) and perpendicular (P) real components of the electric field (with their imaginary counterparts designated by *). The Stokes parameters represent total, linearly polarized, and circularly polarized radiance, which all carry units of W m$^{-2}$ nm$^{-1}$ sr$^{-1}$. The total radiance ($I$) is the sum total of the parallel and perpendicular intensities of the beam. The linearly polarized radiances represent

excesses of 0° over 90° polarization angles ($Q$), and 45° over 135° polarization angles ($U$), and the circularly polarized radiance represents the excess of left-circular over right-circular polarization ($V$). These four parameters fully describe the polarization state of a light beam and are related with two equations:

$$I^2 \geq Q^2 + U^2 + V^2, \tag{6}$$

and

$$DOP = \frac{\sqrt{Q^2 + U^2 + V^2}}{I}, \tag{7}$$

where $DOP$ is the degree of polarization, a dimensionless ratio between 0 and 1 that represents the amount of polarized light in the total intensity measurement. Note that in the absence of V, Eq. (7) becomes the *degree of linear polarization* (*DOLP*). We will neglect the V parameter in this study, as it is negligible at the top of the atmosphere (Hansen and Travis 1974) and not measured by AirHARP.

        Ray traces through optical media, like lenses and prisms, are sequential and can be described by
linear algebra. A polarized beam traveling through an optical interface is related to the output beam by a Mueller matrix:



$$\begin{bmatrix} I \\ Q \\ U \end{bmatrix}_{sca} = \begin{bmatrix} M_{11} & M_{12} & M_{13} \\ M_{21} & M_{22} & M_{23} \\ M_{31} & M_{32} & M_{33} \end{bmatrix} \begin{bmatrix} I \\ Q \\ U \end{bmatrix}_{inc}, \quad (8)$$

where subscripts *inc* and *sca* represent the Stokes vector for the incident beam and scattered beam, respectively. The $M_{ij}$ elements describe how the medium changes the nature this beam. The M-matrix in Eq. (8) may be a single optical element, or an optical train. This matrix is a product of several matrices that describe the sequential optical elements of the AirHARP system:

$$\begin{bmatrix} I \\ Q \\ U \end{bmatrix}_{det} = M_{polarizer}\ M_{prism}\ M_{train} \begin{bmatrix} I \\ Q \\ U \end{bmatrix}_{inc} = \boldsymbol{M_{system}} \begin{bmatrix} I \\ Q \\ U \end{bmatrix}_{inc}, \quad (9)$$

where the subscript *det* now corresponds to the Stokes vector incident on the detector FPA, and the subscripts *polarizer, prism,* and *train* correspond to the Mueller matrices of the detector polarizer, the optical path through the Phillips prism, and the optical lens train in the housing. In theory, each of these

M-matrices defined in Eq. (9) contain internal Mueller matrices for coating interfaces, lenses, and prism elements, but these are difficult to characterize individually from a single full-system detector measurement. Therefore, these are combined into one global M-matrix ($M_{system}$) that characterizes the entire optical train.

The HARP detectors only register intensity values, meaning it is not possible to measure the $Q_{det}$

and $U_{det}$ information directly in Eq. (9). However, because the linear polarizer in front of each detector is oriented at a different angle, the intensity measured at the FPA encodes information about that polarization state. We can retrieve the original polarization state of the Earth scene by combining intensity information from the three detectors (Fernandez-Borda et al. 2009). We can isolate the matrix components from the Eq. (9) matrix that contribute to $I_{det}$, for each detector, and form a relationship between detector

counts and the incident Stokes state:





$$
\begin{bmatrix} DN_{corr,det\ A} \\ DN_{corr,det\ B} \\ DN_{corr,det\ C} \end{bmatrix} = \begin{bmatrix} M_{11,det\ A} & M_{12,det\ A} & M_{13,det\ A} \\ M_{11,det\ B} & M_{12,det\ B} & M_{13,det\ B} \\ M_{11,det\ C} & M_{12,det\ C} & M_{13,det\ C} \end{bmatrix} \begin{bmatrix} I \\ Q \\ U \end{bmatrix}_{inc} = \boldsymbol{M}^* \begin{bmatrix} I \\ Q \\ U \end{bmatrix}_{inc}, \qquad (10)
$$

where the $M_{1j,\ det\ X}$ coefficients represent the first row of the Mueller matrix for the light path through the

optical system into that specific detector (j = 1, 2 or 3) and $DN_{corr,\ det\ X}$ represents the corrected detector counts from Eq. (5), where X could be A, B, or C. This matrix with $M_{1X}$ coefficients is **M\***. Note that **M\*** is not a Mueller matrix.

### 3.4.2 Application in the laboratory

       The purpose of the polarimetric calibration of the AirHARP instrument is to derive **M\*** and/or its

inverse using Eq. (10). To do this, we use an integrating sphere as our source and a 1-inch Moxtek wire-grid linear polarizer, placed at the aperture of this sphere to modify the polarization content of the beam. The Moxtek is a high efficiency, high contrast polarizer suitable for the 400-900nm wavelength range. We set this polarizer in a Thorlabs rotational mount and accurately control the angle of polarization entering the AirHARP instrument to 0.001°. The Moxtek is highly reflective, so we also tilt the polarizer

along the AirHARP optical axis by 10° to avoid back-reflections into the AirHARP optics (van Harten et al. 2018). The polarizer is characterized before any testing and its starting orientation is verified by an external reference polarizer.

       The optical axis of the HARP instrument is placed along the axis between the center of the Moxtek polarizer and the aperture of the integrating sphere such that the HARP image is illuminated at nadir. The

integrating sphere is set to a lamp level below the saturation limit of all HARP channels. The Moxtek is mechanically rotated at intervals of 10°. Simultaneous images are taken at each detector and Moxtek angle. Because we defined the starting orientation of the Moxtek, the relative Stokes state at each angle is well-known, with $Q/I = cos\ 2\vartheta$ and $U/I = sin\ 2\vartheta$ (Kliger et al. 1990), where $\vartheta$ is the rotation angle. It



is important to note that the absolute radiometry is not important at this stage, as long as the source output

is stable.

The optical path from the HARP front lens to a single FPA creates a single partial polarizer (i.e. Eq. 9). Therefore, this test creates a two-polarizer system. Malus' law explains the observed counts at each detector as a function of $\vartheta$. To account for optical complexity of HARP, we use a general fit:

$$DN_{corr,det\ X}(\vartheta) = \alpha\ cos^2[\vartheta - (\vartheta_X - \beta)]\ + \gamma, \tag{11}$$

where the subscript *corr, det X* represents the corrected counts in a single detector (i.e. X could be A, B, or C) during this test and $\alpha, \beta$, and $\gamma$ are fit parameters. $\vartheta_X$ is the nominal polarizer angle for a detector X, determined during AirHARP pre-assembly testing. Figure 6 shows examples of Malus curves for the

three detectors, using co-located Moxtek data at the center of the lens for the 670nm channel.

The amplitude of the curves is related to the $\alpha$ and $\gamma$ parameters, the phase to $\beta$, and the extinction ("lift" off the zero line) to $\gamma$. Any global bias due to the Moxtek polarizer itself is negligible or removable for reasons stated above. Surface inhomogeneities on the polarizer may impart higher-order frequencies in the signal, which can be accounted for by Fourier decomposition (Cairns et al. 1999). A separate

sensitivity study using a reference polarimeter and a rotating polarizer in our lab suggests that Fourier modes at the 0.005 level (such as $sin4\vartheta$) stem from surface variations and are removed during this analysis. After normalizing each Malus curve by the maximum of the curve in *det A* for each channel and detector, and inverting the matrix in Eq. (10), we come to a final relationship that completely represents this step:


$$\begin{bmatrix} 1 \\ -cos\ 2\vartheta \\ sin\ 2\vartheta \end{bmatrix}_{inc} = \begin{bmatrix} C_{11,} & C_{12} & C_{13} \\ C_{21} & C_{22} & C_{23} \\ C_{31} & C_{32} & C_{33} \end{bmatrix} \begin{bmatrix} DN_{corr,det\ A}(\vartheta) \\ DN_{corr,det\ B}(\vartheta) \\ DN_{corr,det\ C}(\vartheta) \end{bmatrix} (max(DN_{corr,det\ A}(\vartheta))^{-1}, \tag{12}$$

where the Stokes parameters (I, Q, U) are replaced with their theoretical forms and the matrix $\mathbf{C} = (\mathbf{M}^*)^{-1}$ from Eq. (10). This $\mathbf{C}$ is defined in Fernandez-Borda et al. (2009) as the *characteristic matrix*. The $\mathbf{C}$



translates normalized, corrected detector counts to normalized Stokes parameters. The **C⁻¹** has an analytic

form based on the angle of the polarizers used for the three detectors (Schott 2009):

$$C^{-1} = \begin{bmatrix} f_A & f_A\,g_A\,cos\,2(\theta_A - \beta_A) & f_A\,g_A\,sin\,2(\theta_A - \beta_A) \\ f_B & f_B\,g_B\,cos\,2(\theta_B - \beta_B) & f_B\,g_B\,sin\,2(\theta_B - \beta_B) \\ f_C & f_C\,g_C\,cos\,2(\theta_C - \beta_C) & f_C\,g_C\,sin\,2(\theta_C - \beta_C) \end{bmatrix}, \tag{13}$$

Coefficients define the transmission of the light through the entire optical system ($f_X$), polarizing

efficiency ($g_X$), and phase offset ($\beta_X$) relative to the nominal detector polarizer angles ($\theta_X$) from Eq. (11).

This characteristic matrix can be solved in two ways: a least-squares approach on Eq. (12) using data

from at least three Moxtek polarizer angles, or a similar least-squares approach on the inverse matrix, Eq.

(13). We prefer the former in this study, but consistency checks with the latter are useful. Table 2a gives

the characteristic matrix coefficients with relative uncertainties using the least-squares method and Table

2b gives example values with uncertainties using the parametric method. Both tables shown below

represent a 4x4 nadir pixel bin for the 670nm channel for AirHARP.

        Table 2b shows that the nominal AirHARP polarizer angles ($\theta_X$) can derivate from their expected

values ($\beta_X$). Note that $\theta_X - \beta_X$ is the perceived polarization orientation of the entire light path from the

perspective of each FPA. Retardances induced by the prism and/or detector polarizer will contribute to

$\beta_X$. Note that the coefficients are significantly different from the Pickering matrix, the ideal C-matrix for

a AirHARP-like system (Schott 2009). The characteristic matrix coefficients shown in Table 2a use the

polarizer datasets alone, though current AirHARP L1B processing through Version 002 includes input

from low DOLP sources (integrating spheres, partial polarization generators) for closure on the entire





DOLP range. The errors and values in Table 2a can be used to calculate the propagated uncertainty in the relative Stokes parameters, which is derived from Eq. (12):

$$\sigma_{S_j}{}^2 = (max(DN_{corr,det\,A}(\vartheta))^{-2} \sum_{i=1}^{3} \left[ \left( DN_{corr,i}\,\sigma_{C_{ij}} \right)^2 + \left( C_{ij}\,\sigma_{DN_{corr,i}} \right)^2 \right], \qquad (14)$$

where $\sigma_{S_j}$ is the 1-sigma standard deviation of the Stokes parameters (denoted generally by subscript $S$). We use the $j$ iterant to define the Stokes parameter: [1,2,3] corresponds to [I,Q,U], and can be used interchangably. $DN_{corr,i}$ is the result from Eq. (5) where the $i$ iterant [1,2,3] corresponds to sensors [A, B, C]. $\sigma_{C_{ij}}$ is the uncertainty quoted in Table 2b for the $C_{ij}$ matrix element, and $\sigma_{DN_{corr,i}}$ is the propagated uncertainty of the detector counts measurement. The value for $\sigma_{DN_{corr,i}}$ involves random elements such

as shot, read, and dark current noises, and systematic elements from background, flatfield, and non-linear correction. It may also include stray light and other noises that are difficult to decouple. At the integration times we use, shot noise dominates, so  a 1-sigma standard deviation of data from a real AirHARP superpixel is used.

## 3.5 Radiometric calibration

### 3.5.1 Relative spectral response

With the polarimetric calibration complete, the next step is radiometric calibration, which requires knowledge of spectral response. The AirHARP instrument uses several filters to define the four nominal wavelength channels, with bandwidths in parentheses: 440 (16), 550 (13), 670 (18), and 870 (39) nm. The SRF is defined by a multi-bandpass filter (MBPF) and the stripe filter on top of each detector.

To validate these filter specs, we placed the AirHARP instrument in the aperture of a separate 25.6cm integrating sphere at NASA GSFC, fed by an Ekspla laser source. The Ekspla is a scanning monochromator capable of 1 nm precision, over a 200-1000nm range. We set the Ekspla source at a given wavelength and verified each output channel and bandwidth using an external Avantes spectrometer. We





use the spectrometer output to correct the AirHARP measurements for any variation in Ekspla laser power

over the course of the testing period.

The standard process is used on AirHARP images that are taken at each Ekspla wavelength setting. The Ekspla channels were chosen using *a priori* knowledge of the filter spectra from the manufacturer. A higher density of images were acquired in-band than out-of-band to capture the structure of the in-band SRF. Figure 7a shows AirHARP images of the integrating sphere, illuminated by four in-band Ekspla

wavelengths. When the Ekspla is set to an in-band channel near 670nm, the 60 AirHARP red view sectors are illuminated. For the other AirHARP channels, the sparser distribution of 20 view sectors appear whenever the Ekpsla is in-band. For Ekpsla wavelengths rejected by the AirHARP system, the images are compatible with dark signal (Figure 2b).

Using the telecentric technique, we take a small region of nadir pixels, correct their values via the

process leading up to Eq. (5), and plot them against Ekspla wavelength for a single HARP channel. Figure 7b shows the SRF for AirHARP Sensor 1 (blue dots), Sensor 2 (green dots), and Sensor 3 (orange dots) for 440nm (left), 550nm (left-center), 670nm (right-center), and 870nm (right). Because the SRF data is noisy, even after correction from an external spectrometer, we use a general super-Gaussian fit of order 6 (plotted in gray) to simplify the following analysis. Note that the edges of the in-band response are well-

defined, and the 870nm channel shows the aggressive narrowing of the leading edge of the SRF as discussed earlier. Figure 7b also shows a differential SRF for the AirHARP 440nm band, which is likely due to manufacturer error in the thin-film coating for the AirHARP prism interfaces or detector stripe filters. This difference may impact AirHARP 440nm L1B radiances in field data that is calculated using lab-derived coefficients. Because Rayleigh scattering is so strong in the neighborhood of 440 nm, offsets

in the L1B data can be treated as an increase in stray light, which correlates with the signal (and therefore the SRF). A simple Rayleigh-like SRF adjustment in each sensor may correct for this SRF differential. This 440nm SRF differential is unique to AirHARP; we see no evidence of this in the HARP CubeSat or HARP2 440nm designs. Further vicarious corrections on AirHARP 440nm L1B data are underway.



This testing benefits two studies: (1) calculation of extraterrestrial solar irradiance, used to convert radiance measured at TOA to reflectance, and (2) radiometric calibration. To perform (1), we integrate the solar spectrum (here, using the American Society for Testing and Materials Standard Extraterrestrial Spectrum Reference E-490 Air Mass Zero (AMZ) database (NREL 2000) inside the SRF for each HARP wavelength:

$$F_0(\lambda) = \frac{1}{\Delta\lambda} \int_{\lambda_i}^{\lambda_f} B(\lambda)\, SRF(\lambda)\, d\lambda, \tag{15}$$

where $\lambda$ is the wavelength (subscripts $i$ and $f$ denoting the trailing and leading edges of the spectral band) in nm, $\Delta\lambda$ is the bandwidth in nm, $B(\lambda)$ is the solar spectral irradiance in W m$^{-2}$ nm$^{-1}$ and SRF($\lambda$) is the spectral response function. We only use the structure of the in-band channel in Eq. (15), and fit each window to a 6$^{th}$ order super-Gaussian function, due to unexplained noise in the dataset larger than the uncertainty of each data point (especially at 870nm). Normalized out-of-band rejection is at or below 0.001 for the 300 to 1050 nm range, as well. Analysis of the second-order in-band differences relative to this theoretical fitting are ongoing but are not expected to contribute significantly to the L1B data product (AirHARP 440nm notwithstanding). Table 3 shows the details of our spectral response testing and the extraterrestrial solar irradiance, $F_0$, calculated using Eq. (16), for each channel.

The final column of this chart is used to convert measured radiances to reflectances as per:

$$\rho(\lambda) \;=\; \frac{\pi\, L(\lambda)}{F_0(\lambda)}, \tag{16}$$





where $\rho(\lambda)$ is the reflectance and $L(\lambda)$ is radiance in units of W m$^{-2}$ nm$^{-1}$ sr$^{-1}$, assuming a Lambertian scattering distribution of light in the pixel. It is convention to sometimes include an extra term in the denominator of Eq. (17) to account for the view zenith angle, but it is omitted here to remain consistent with later discussions.

### 3.5.2 Gain characterization

Our radiometric calibration translates the normalized Stokes parameters to calibrated radiances (W m$^{-2}$ nm$^{-1}$ sr$^{-1}$). This step gives scientific weight to our measurements and allows us to retrieve radiative properties about the atmosphere and surface. Again, integrating spheres are optimal for this testing. For example, the radiometrically calibrated NASA GSFC "Grande" sphere is traceable to the National Institute of Standards and Technology (NIST) with calibration uncertainties publicly available for a comparable sphere (Cooper and Butler 2020). The spectral sensitivity of "Grande" peaks around 1 $\mu$m and the illumination of the nine lamps is linear.

We set the AirHARP instrument in the same conditions as the polarimetric calibration, discussed in 1.4, except with no polarizing element between the instrument and integrating sphere. Because the lamps are incandescent sources, we adjust the AirHARP detector integration times to capture enough signal in the blue channel and stay out of saturation in the NIR. The standard process is used at each lamp level to create template images. Using the telecentric technique, we select a small nadir pixel bin for a given wavelength, correct the values using the process leading up to Eq. (4), and apply the characteristic matrix for that channel to the co-located data in each detector. The sphere output is depolarized, so the resulting Stokes parameters Q and U are statistically zero and the total intensity, I, contains all of the information content. As per Eq. (12), the resulting I is in counts, yet represents the band-weighted signal measured by a particular AirHARP channel. To find the equivalent radiance levels as observed by AirHARP, the solar spectrum, B($\lambda$), is replaced by the Grande SRF in Eq. (16) (Cooper and Butler 2020) and this calculation is performed for each lamp and wavelength. The radiometric calibration derives the





slope (W m$^{-2}$ nm$^{-1}$ sr$^{-1}$ ADU$^{-1}$) that translates the normalized AirHARP intensities to the calibrated radiances:

$$L_{lamp} = k \left( C_{11} DN_{corr,det\ A} + C_{12} DN_{corr,det\ B} + C_{13} DN_{corr,det\ C} \right) + \gamma, \tag{17}$$

where $L_{lamp}$ is the calibrated irradiances at that lamp level. $k$ is our gain factor, and $\gamma$ is a linear bias. The equivalent Grande radiances and measured counts share the same pixel solid angle as observed by AirHARP, so this is implicitly factored into Eq. (17). For all channels, the linear bias $\gamma$ is compatible with
zero within 3-sigma. Therefore, the general calibration equation for the AirHARP instrument is the following:

$$\begin{bmatrix} I \\ Q \\ U \end{bmatrix} = k \begin{bmatrix} C_{11} & C_{12} & C_{13} \\ C_{21} & C_{22} & C_{23} \\ C_{31} & C_{32} & C_{33} \end{bmatrix} \begin{bmatrix} DN_{corr,det\ A} \\ DN_{corr,det\ B} \\ DN_{corr,det\ C} \end{bmatrix}, \tag{18}$$

and the complete, propagated uncertainty of the L1B calibrated radiances:

$$\sigma_{S_j}{}^2 = \sum_{i=1}^{3} k^2 \left[ \left( DN_{corr,i}\ \sigma_{C_{ij}} \right)^2 + \left( C_{ij}\ \sigma_{DN_{corr,i}} \right)^2 \right] + \left( C_{ij}\ DN_{corr,i}\ \sigma_k \right)^2, \tag{19}$$

where the subscripts follow the same convention as Eq. (14). Note that if the signal-to-noise ratio of each
detector is preferred, $\sigma_{DN_{corr,i}}$ can be substituted for $DN_{corr,i}\ SNR_i^{-1}$ and further consolidation of terms is possible.



## 4 Validation of calibrated measurements

### 4.1 Nadir coefficients

Before we evaluate the calibration over the entire FOV, it is important we validate the same lens
locations that we used to calibrate the instrument. Here, we evaluate the nadir coefficients for a range of
partially polarized DOLP signals, similar to those AirHARP observes in field data.

In the atmosphere, DOLP measurements close to 1 occur only at certain geometries with sunglint
over dark ocean or Rayleigh scattering in the ultraviolet. More often, a complex atmosphere-land-ocean
scene generates partially polarized light ($0 < \mathrm{DOLP} < 1$). To simulate this, a partial polarization generator
box (POLBOX), a Fresnel device comprised of two rotatable glass blades at equal index of refraction, is
used (Figure 8a). This polarization state generator is widely used for lab validation of spaceborne
polarimeters (van Harten et al. 2018, Li et al. 2018, Smit et al. 2019). This POLBOX system is unique in
that it *conserves DOLP*: when the entire POLBOX system rotates along its center axis, any DOLP
measured at fixed blade angle remains the same, while the absolute values of Q and U will change. At a
fixed global rotation angle, steeper inclinations of the glass blades will increase the DOLP up to ~60%,
due a rotation limit of the blades. Any deviation in the DOLP retrieval gives the lab calibration uncertainty
of the HARP system, after systematic POLBOX uncertainty is accounted for. The POLBOX DOLP is
analytic and the values at each blade setting can be determined by the sequential Fresnel interactions at
each air-glass interface:


$$DoLP_{POLBOX} = \frac{\alpha(n,\lambda)\cos^2(2\theta) + \beta(n,\lambda)\cos(2\theta) + \gamma(n,\lambda)}{\varepsilon(n,\lambda)\cos^2(2\theta) + \mu(n,\lambda)\cos(2\theta) + \omega(n,\lambda)} \qquad (20)$$

where $\alpha, \beta, \gamma, \varepsilon, \mu,$ and $\omega$ are glass-specific coefficients, dependent on refractive index, $n$, and
wavelength, $\lambda$, and $\theta$ is the glass blade angle. For our test, we keep the POLBOX glass blades
perpendicular to the table, and take HARP images at increasing blade angles. The angle of the blades is
controlled by a fine micrometer dial, and the angle is known within 0.25°. The data is corrected through
the process leading up to Eq. (4), and the pre-computed calibration matrices are applied for each





wavelength and image in the dataset. As mentioned above, the characteristic matrix used in this validation includes Moxtek polarizer data and input Stokes vectors that represent unpolarized light for closure over

the entire DOLP range. Using the same nadir pixel bin that was used for calibration, the measured Stokes parameters at each POLBOX blade angle are processed into the DOLP via Eq. (8) and these results are compared to Eq. (20) for each blade angle and wavelength.

The measured DOLP from the HARP system is within ±0.5% (RMS 0.25%) of the true POLBOX values for all wavelengths, given a 4x4 pixel nadir bin, as shown in Figure 8b-c. Glass blade angles (<5˚)

that create back reflections in the wide HARP FOV are neglected from the comparison. Removing these angles has a negligble impact on the comparison, as the theoretical DOLP at 10˚ is still quite low (~4%) and still represents a depolarized environment. The POLBOX itself imparts a static DOLP uncertainty of 0.0015, related to the uncertainty in the glass blade angle (Li et al. 2018). This experiment is only limited by the intensity of the integrating sphere, which here was no less than 0.09 in reflectance (440nm). This

level is a bit higher than the typical aerosol signal used in theoretical experiments ($L_{typ}$), but it is challenging to balance integration time and saturation in a single lab measurement when all channels are simultaneously exposed. Even so, we conclude that the HARP design allows for a highly accurate pre-launch DOLP baseline for all channels, relative to recommended cloud and aerosol science uncertainty benchmarks (NASA 2015, NASA 2021). POLBOX testing at different levels of reflectance are

anticipated during the HARP2 pre-launch baseline testing in late 2022 and a comprehensive error model for the HARP instruments is anticipated in future work.

## 4.2 Full FOV intercomparisons with field data

## 4.2.1 AirHARP participation in the Aerosol Characterization with Polarimeter and LIDAR
(ACEPOL) campaign

Sensitivity tests in the lab allow us to characterize the HARP instrument in a well-controlled setting. However, these environments can be limited by resources and time, and this can impact how much of the FOV, spectral channels, and dynamic range are characterized. To validate the full FOV calibration, we



take field data and compare how the HARP instrument measures the multi-angle reflectance and polarized
signal with a similar MAP over a common target.

AirHARP participated in two NASA aircraft campaigns in 2017: the Lake Michigan Ozone Study (LMOS) and Aerosol Characterization from Polarimeter and Lidar (ACEPOL). LMOS took place over Lake Michigan and eastern Wisconsin from May 25 to June 19 2017 and ACEPOL over the southwestern United States and eastern Pacific Ocean from October 23 to November 9 2017. LMOS was AirHARP's debut and was the only instrument of its kind taking measurements during this period. ACEPOL, on the other hand, included two lidar and four polarimeter instruments on the aircraft, including AirHARP. A major goal of the ACEPOL campaign was to compare different polarimeter concepts over common targets, improve cross-calibration studies, and develop new synergistic algorithms for retrieving aerosol, cloud, land, and ocean properties.

During ACEPOL, these six instruments observed over 30 scenes including urban cities, coastal oceans, dry lakes, cloud decks, and prescribed wildfire smoke. Two of these targets are best suited for reflective solar band calibration and validation: sunglint over dark ocean and the Rosamond Dry Lake, a flat desert site in California. Sunglint is highly polarized at some geometries, reaching DOLP of nearly 1 in the optical regime. Off-glint, polarization is reduced and low ocean albedo is useful to validate dim reflectances. The sunglint signal can be modeled accurately, if the viewing and solar geometry are known and aerosol and Rayleigh scattering are removed. The appearance of sunglint depends on the ocean surface wind speed, which can roughen the surface and break up the signal (Cox and Munk 1954). Even despite strong surface winds, the ocean surface is considered flat from a viewing altitude of 20km and requires no special topography correction to the data. Multi-angle polarimeters, like AirHARP, measure the way the sunglint signal varies with viewing angle, and can reproduce a discrete intensity and polarization profile with angle. Therefore, sunglint datasets are very convenient to use for calibration validation. The Rosamond Dry Lake is also a useful calibration target: it is a pseudo-invariant, highly reflective surface with a low DOLP profile. We will use two sunglint (10/23/2020) and two Rosamond





Dry Lake (10/25/2020) scenes from the campaign to show that our telecentric technique captures the
expected performance of the AirHARP instrument across the FOV.

Because the focus of this work is calibration and not data intercomparisons, we will present the
following study in a simple and limited sense. The Research Scanning Polarimeter (RSP) instrument was
chosen as our validator because it best matches the along-track angular sampling of HARP, shared the
same wing of the ER-2 with AirHARP during ACEPOL, and has the longest history of accurate, validated
polarimetric measurements. The following describes the process used to co-locate AirHARP and RSP
measurements at similar viewing angles:

1. A target of interest and reference lat-lon pair is identified and the closest scan in the RSP data is
   found. The average lat-lon pair of this scan becomes the new reference lat-lon point.
2. The algorithm finds the closest matching view zenith angles between AirHARP and RSP within
   0.5° over this common target.
3. For each matching view angle, an 8x8 pixel search window is defined in the AirHARP granule
   around the lat-lon for that RSP view angle. Using each pixel in the search window as a new target,
   the AirHARP data is binned 8x8 around this pixel to approximate the RSP 220 m ground
resolution. A cost function is used to calculate the error-normalized difference between the
   AirHARP-RSB measurements for that view angle. The closest matching AirHARP superpixel will
   minimize the cost function.
4. The angular data of the RSP is interpolated to the exact scattering angle range measured by
   AirHARP.
5. This process is repeated for all relevant spectral channels.

We use a search method in this comparison to further reduce the differences in pointing between the two
instruments, which can be complicated by wing flex, differences in inertial monitoring, vibrations and





thermal changes, and geolocation references used between both science teams. The cost function used to
minimize the closest match between the two instruments:

$$\chi^2 = \frac{(R_{RSP} - R_{AirHARP})^2}{(\Delta R)_{RSP}^2 - (\Delta R)_{AirHARP}^2} + \frac{(P_{RSP} - P_{AirHARP})^2}{(\Delta P)_{RSP}^2 - (\Delta P)_{AirHARP}^2},$$  (21)

where $R$ and $P$ are total and polarized reflectances (i.e. the Stokes parameter I and $\sqrt{Q^2 + U^2}$,
respectively). The delta terms represent the measurement uncertainty using the error model for the
respective instrument. The RSP model is given in the Appendix and the AirHARP model is given in the
next section. This cost function is robust against scenes with a notable differences in DOLP and
reflectance. All products used in Eq. (21) correspond to the same location on the ground, as determined
by the above steps. Because this paper uses these intercomparisons to prove our full FOV calibration, we
only need a few datasets over different viewing and solar geometries to make our case. The following
will discuss the results of the AirHARP and RSP intercomparison over both ocean and desert sites during
the ACEPOL campaign.

**Section 4.2.2 Results and Discussion**


The full FOV comparison with RSP uses two ocean cases from October 23, 2020 and two desert cases
from October 25, 2020, taken during the ACEPOL campaign. The ocean captures occurred 30 minutes
apart off the coast of California: the first at 20:10 UTC over 35.12° N 124.75° W and the second at 20:49
UTC over 31.75° N 122.38° W. We will identify the earlier as Ocean 1 and the later as Ocean 2 going
forward, and both are parallel to and slightly off the solar principal plane. The desert cases were taken on
October 25, 2020 over the Rosamond Dry Lake site in California, also 30 minutes apart: the first at 17:28
UTC and the second at 17:57 UTC. These captures will be identified as Desert 1 and Desert 2, and both
targeted the general region around 34.83° N 118.07° W.

The AirHARP and RSP data were ordered for these dates, times, and locations, and the co-location
procedure described in Section 3.2.1 was followed for each of the sites and the three spectral channels in



common to both instruments: 550, 670, and 865 or 870nm. We do not show a comparison with the AirHARP 440nm band because there is no comparable RSP channel and for SRF reasons mentioned above that could complicate the interpretation of the results. The AirHARP 550 (13), 670 (18), and 870 (39) nm and the RSP 550 (20), 670 (20), and 865 (20) nm spectral bands are generally compatible. We also do not expect any significant differences in the signal of the desert or glint targets relative to SRF differences. We did not perform any spectral matching in this work.

Figure 9 shows a multi-angle co-located comparison AirHARP and RSP for the four ACEPOL datasets. RSP data is in black and the AirHARP desert (red) and ocean (blue) for both reflectance (first column) and DOLP (second column). Three compatible channels are shown: 550nm (top row), 670nm (middle row), and 870nm (bottom row). The error bar on the AirHARP points is the sub-pixel standard deviation of the superpixel at each angle. For the ocean cases, the reflectance is lower than 0.1 in all channels, but the DOLP range is wide, 0 to ~0.8. The desert cases were chosen specifically to contrast with sunglint. The desert cases represent the same target viewed from two different headings. The dependency on viewing geometry is clear in the separation of the desert reflectance curves in all channels. These cases provide a range of geometry for intercomparison and adequate contrast in reflectance and DOLP to validate our calibration.

Figure 10 shows a one-to-one comparison of co-located AirHARP and RSP reflectance and DOLP, across their three common spectral bands for the above ACEPOL datasets. The plots are log-scaled to show differences for dim reflectance and low DOLP. The AirHARP data reproduces the global structure of the RSP for all sites in all channels. The RSP uncertainty is calculated using their error model and *a priori* inputs to reflectance and DOLP equations, and the AirHARP uncertainty model leverages superpixel statistics (see Appendix). The instruments agree within 1% in reflectance and DOLP for most VZA up to ~35° for all three channels. This difference range is reasonable given similar, recent studies (Knobelspiesse et al. 2019, Smit et al. 2019, van Harten et al. 2018). Beyond ~35° VZA, there is a systematic difference between AirHARP and RSP, which may be tied to a variety of error sources.

First, AirHARP did not have an on-board calibrator, mechanism of temperature regulation, or dry purge during ACEPOL. If the field measurement was impacted by ascent-descent humidity changes, differences in temperature between the aircraft pod and the outside environment, or condensation of water



and aggregation of ice particles on the front lens, these effects may be difficult to characterize. These may
have asymmetric impacts on the data at different FOVs as well.

Specifically, Figure 9 shows some deviations between the AirHARP-RSP measurements, especially at larger scattering angles at 670nm and 870nm. This deviation may also be connected to georegistration at the widest angles, any unaccounted for misregistration between RSP and AirHARP, and/or interpolation at the AirHARP L1B stage. The HARP front lens distorts the ground projection by a
factor of 4 at the furthest angles relative to nadir, so the amount of interpolation needed to fit the data on a common L1B grid is much more intense at far angles. This is complicated by "pitch surfing" of the ER-2. In several cases during ACEPOL, the ER-2 hit slight turbulence during flight, which briefly tilted the AirHARP instrument off-nadir. Pitch surfing may grow the pixel projection at far angles and adds uncertainty in our interpolation of these angles in the gridding stage. The wider deviant angles in the
Figure 9 comparison are desert cases (red in Figure 9). If RSP-AirHARP misregistration is a major factor, the angular signals from the ocean may be more robust against slight changes in pointing than desert surfaces. Figure 9 does not show the same differences in the ocean scenes (blue) for these angles and channels.

However, the overall structure of the RSP signal is reproduced by AirHARP instrument across two
different scenes, a wide range of view angles, and within statistical uncertainty for most values. These results show the strength of our simple and efficient telecentric technique. The accuracy of this calibration is indirectly demonstrated in AirHARP Level 2 aerosol and cloud retrieval studies in Puthukuddy et al. (2020) and McBride et al. (2020), as well. Both of these studies use full-FOV datasets.

It is also important to note that cross-validation between instruments cannot determine which
instrument is "more correct", only how well they both agree over a common range of angles, channels, and targets. The community anticipates a third-party intercomparison study in the future that compares the measurements of all ACEPOL polarimeters with each other, vector radiative transfer models, and other co-located satellite instruments.



# 5 Conclusions

The AirHARP calibration pipeline presented in this work exceeds the community requirement of 0.5% DOLP in the lab and reproduces the signal of natural targets relative to another co-located polarimeter. The telecentric calibration scheme is as effective as it is simple. It is also possible in a variety of environments: in space, where physical access is impossible, and during field campaigns, where time and access to the instrument is limited. If a flatfield measurement is done regularly and consistently, the

performance of the entire FPA can be traced through a range of temperatures and humidity environments (on aircraft). The upcoming HARP2 instrument on the NASA PACE mission will include an internal calibrator to validate the full FOV performance throughout the life of the mission.

        The telecentric technique can be used for vicarious calibration with field data alone, too. In the lab calibration, we used a rotating polarizer-sphere setup and pixels at the center of the lens to calculate the

characteristic matrix. This is a special case. In general, any polarized target viewed from at least three different angles may provide enough information to trend the characteristic matrix. It is important that the target is viewed from significantly different geometry (optimally with views parallel and perpendicular to the solar plane, and/or at least three attack angles 60° apart). This achieves the highest discrimination between polarization states (Tyo et al. 2006). Therefore, sunglint, cloudbow, dry lake, salt flat, aerosol

plume, polar ice, and other natural targets can be excellent homogeneous and/or stable vicarious calibration targets. Measurements of these targets, combined with an internal flatfield measurement, may allow for an effective and efficient trending of the instrument.

        The telecentric technique can also be used to cross-calibrate HARP against other polarimetric instruments. For example, any bias in the comparison of reflectance in Figure 10 could generate a

radiometric correction factor that could be applied to the characteristic matrix in Eq. (20). Because the radiometric *k-factor* applies to the entire matrix, a single co-located intercomparison between like instruments is enough to correct the measurement. Using co-located instruments in this way also transfers their uncertainty in geolocation, measurement accuracy, and pointing. Nevertheless, it is invaluable over ill-modeled targets and/or validating against solar or lunar views. The HARP science team is currently

evaluating how this telecentric technique can improve the in-flight calibration of AirHARP and HARP



CubeSat data. We anticipate these methods will be used during the HARP2 deployment in 2024 and beyond.





## 6 Appendix

The RSP error model is provided in Knobelspiesse (2015). The overall error in reflectance (in the above
text defined as R and   and DOLP is described below:

$$\sigma_\rho^2 = \left(\frac{r^2 \sigma_{floor}}{\mu_s}\right)^2 + \frac{a\, R_I r^2}{\mu_s} + \frac{\sigma_{lnK}^2 R_P^2}{16} + \sigma_{a_c}^2 R_I^2 \tag{22}$$

$$\sigma_{DOLP}^2(noise) = 4\left(1 + \frac{DOLP^2}{2}\right)\left(\frac{r^2 \sigma_{floor}}{\mu_s R_I}\right)^2 + 2\left(1 - \frac{DOLP^2}{2}\right)\left(\frac{a\, r^2}{\mu_s R_I}\right) \tag{23}$$


$$\sigma_{DOLP}^2(cal) = \frac{\sigma_{lnK}^2}{2}\left[1 - DOLP^2 + \frac{DOLP^4}{2}\left(1 - \frac{1}{2}\sin^2 4\chi\right)\right] + \sigma_{lna}^2 DOLP^2 \tag{24}$$

$$\sigma_{DOLP}^2 = \sigma_{DOLP}^2(noise) + \sigma_{DOLP}^2(cal) \tag{25}$$

Several parameters are prescribed, based on Knobelspiesse (2015):

- Solar distance in AU, r: 1
- Noise floor, $\sigma_{floor}$ (x $10^{-5}$): 2.5 (550 nm), 2.2 (670 nm), and 2.0 (865 nm)
- Shot noise parameter, $a$ (x $10^{-9}$): 4.5 (550 nm), 3.7 (670 nm), 3.7 (865 nm)
- Relative gain coefficient cal uncertainty, $\sigma_{lnK}$: 0.005
- Absolute radiometric uncertainty, $\sigma_{a_c}$: 0.03
- Polarimetric characterization uncertainty, $\sigma_{lna}$: 0.001

Other parameters are given in the field datasets and are a function of observational geometry and Earth
scene:

- Cosine of the solar zenith angle, $\mu_s$
- Intensity reflectance, $R_I$



- Polarized reflectance, $R_P$
- Degree of Linear Polarization, $DOLP$

Finally, the RSP DOLP uncertainty depends on the angle of polarization, $\chi$, in Eq. (24). In a sensitivity
study with the above parameters and field data, we found that the intercomparison with AirHARP did not



vary meaningfully when $\chi$ varied between 0 and 180°. Therefore, $\sin^2 4\chi$ was set to its expectation value, 0.5, which represents any angle $\chi = (45n + 11.25)\,°$ for $n$ in $\mathbb{Z}$.

The simplified AirHARP error model is described below:

$$\left(\frac{\Delta\rho}{\rho}\right)^2 = (0.03)^2 + \frac{1}{B}\left(\frac{\sigma_\rho}{\rho}\right)^2 \tag{26}$$

$$\Delta DOLP^2 = (0.0025)^2 + \frac{1}{B}\sigma^2_{DOLP} \tag{27}$$


where $\sigma$ (for reflectance or DOLP) the 1-sigma standard deviation of the AirHARP superpixel. The $B$ corresponds to the number of binned pixels.

## 7 Data Availability

NASA ACEPOL L1B datasets are available on https://www-air.larc.nasa.gov/cgi-bin/ArcView/acepol under "MARTINS J. VANDERLEI" (AirHARP) and "CAIRNS, BRIAN" (RSP). The specific AirHARP datasets used in this work are:

ACEPOL-AIRHARP-L1B_ER2_20171023201049_R2.h5 (Ocean 1)
ACEPOL-AIRHARP-L1B_ER2_20171023204956_R2.h5 (Ocean 2)
ACEPOL-AIRHARP-L1B_ER2_20171025172822_R2.h5 (Desert 1)
ACEPOL-AIRHARP-L1B_ER2_20171025175722_R2.h5 (Desert 2)

The RSP datasets used are:


ACEPOL-RSP2-L1B_ER2_20171023195451_R0.h5 (Ocean 1)



ACEPOL-RSP2-L1B_ER2_20171023204417_R0.h5 (Ocean 2)

ACEPOL-RSP2-L1B_ER2_20171025171811_R0.h5 (Desert 1)

ACEPOL-RSP2-L1B_ER2_20171025175031_R0.h5 (Desert 2)


RSP data from ACEPOL and other field campaigns is also publicly available at https://data.giss.nasa.gov/pub/rsp/data/. The AirHARP pre-launch calibration data and codes are available on request from the corresponding author.

## 7 Author contributions

BM developed the testing procedure, performed all lab and field calibrations for AirHARP, performed the above analysis, and wrote this manuscript. BM and HMBJ led the AirHARP participation in the field during the ACEPOL campaign. JVM is the principal investigator of the HARP missions. JVM, JDC, and RFB led the engineering design and development, and supported AirHARP in the field with BM and HMBJ. AP, XX, and NS provided intellectual contributions to several sections, from the perspective of

HARP CubeSat on-orbit data. BC is the principal investigator for the RSP instrument and data contact. All co-authors made substantial edits and reviews to the manuscript.

## 8 Competing Interests

The authors declare no conflicts of interest.

## 9 Disclaimer

The statements contained within the research article are not the opinions of the funding agency or the U.S. government, but reflect the author's opinions.

## 10 Acknowledgements

The authors thank the engineers and support staff at the UMBC Earth and Space Institute for their continued support of the AirHARP, HARP CubeSat, and HARP2 missions. We also acknowledge the



ER-2 support personnel at the NASA Armstrong Flight Center during the NASA ACEPOL campaign,
       especially the pilots who graciously operated AirHARP in targeting mode on many flights. We also thank
       Jim Butler and John Cooper at the NASA GSFC Radiometric Calibration Facility for assisting with the
       many pre-launch characterizations of the HARP instruments. Finally, the authors also thank Pengwang
       Zhai, Martjin Smit, Kirk Knobelspiesse, Samuel Pellicori, Peter Dogoda, and Lorraine Remer for their
contributions to and insights on this work over the years.

**11 Financial support**

       Brent A. McBride received funding from the NASA Earth and Space Science Fellowship (grant no. 18-
       EARTH18R-40) under the NASA Science Mission Directorate. This study is supported and monitored
       by the National Oceanic and Atmospheric Administration – Cooperative Science Center for Earth System
Sciences and Remote Sensing Technologies (NOAA-CESSRST) (grant no. NA16SEC4810008). Brent
       A. McBride has been supported by the City College of New York, the NOAA-CESSRT program, and
       NOAA Office of Education (Educational Partnership Program). ACEPOL flight hours were funded in
       part by the SRON Netherlands Institute for Space Research and the NWO/NSO under project number
       ALW-GO/16-09.




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



## 11 Tables

Table 1. Specifications of the AirHARP detectors

| Number of Active Pixels | 2048x2048 |
|---|---|
| Pixel Size | 7.4 x 7.4 µm |
| Quantum Efficiency (440,550,670,870 nm) | 0.52, 0.50, 0.31, 0.07 |
| RMS Read Noise | 12 e$^-$ |
| Dark Current | 3 e$^-$/s |
| Operational Integration Time | 20 ms |

Table 2a. Example characteristic matrix elements, $C_{ij}$, for the 670nm AirHARP band, via Eq. (12).

| $C_{ij}$ | j = 1 | j = 2 | j = 3 |
|---|---|---|---|
| i = 1 | 1.020 ± 0.001 | -0.053 ± 0.002 | 0.848 ± <0.001 |
| i = 2 | -0.843 ± 0.001 | -0.309 ± 0.001 | 0.938 ± <0.001 |
| i = 3 | -1.257 ± <0.001 | 2.230 ± 0.001 | -0.689 ± <0.001 |

Table 2b. Example of instrument-relative parameters for 670nm AirHARP band, via Eq. (13).

| | f (%/100) | g (%/100) | $\beta$(°) |
|---|---|---|---|
| Sensor A ($\theta_A = 0°$) | 0.501 ± <0.001 | 0.994 ± 0.002 | -3.261 ± 0.060 |
| Sensor B ($\theta_B = 45°$) | 0.471 ± <0.001 | 0.970 ± 0.002 | -6.115 ± 0.048 |
| Sensor C ($\theta_C = 90°$) | 0.605 ± <0.001 | 0.985 ± 0.003 | -4.608 ± 0.060 |

Table 3. Derived AirHARP parameters from spectral response analysis

| Nominal Channel (nm) | Center (nm) | Bandwidth (nm) | $F_0$ (W m$^{-2}$ nm$^{-1}$) |
|---|---|---|---|
| 440nm | 441.4 | 15.7 | 1.855 |
| 550nm | 549.8 | 12.4 | 1.873 |
| 670nm | 669.4 | 18.1 | 1.534 |
| 870nm | 867.8 | 38.7 | 0.965 |





## 12 Figures and Captions


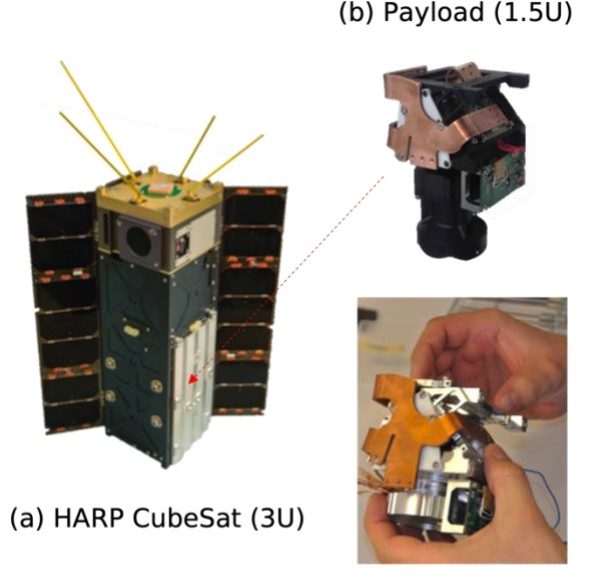

**Figure 1.** AirHARP is an aircraft demonstration of the HARP CubeSat (a), a standalone 3U spacecraft, which carries the same the 1.5U instrument (b) in the lower half of the housing. The payload can fit in the palm of a human hand (c).






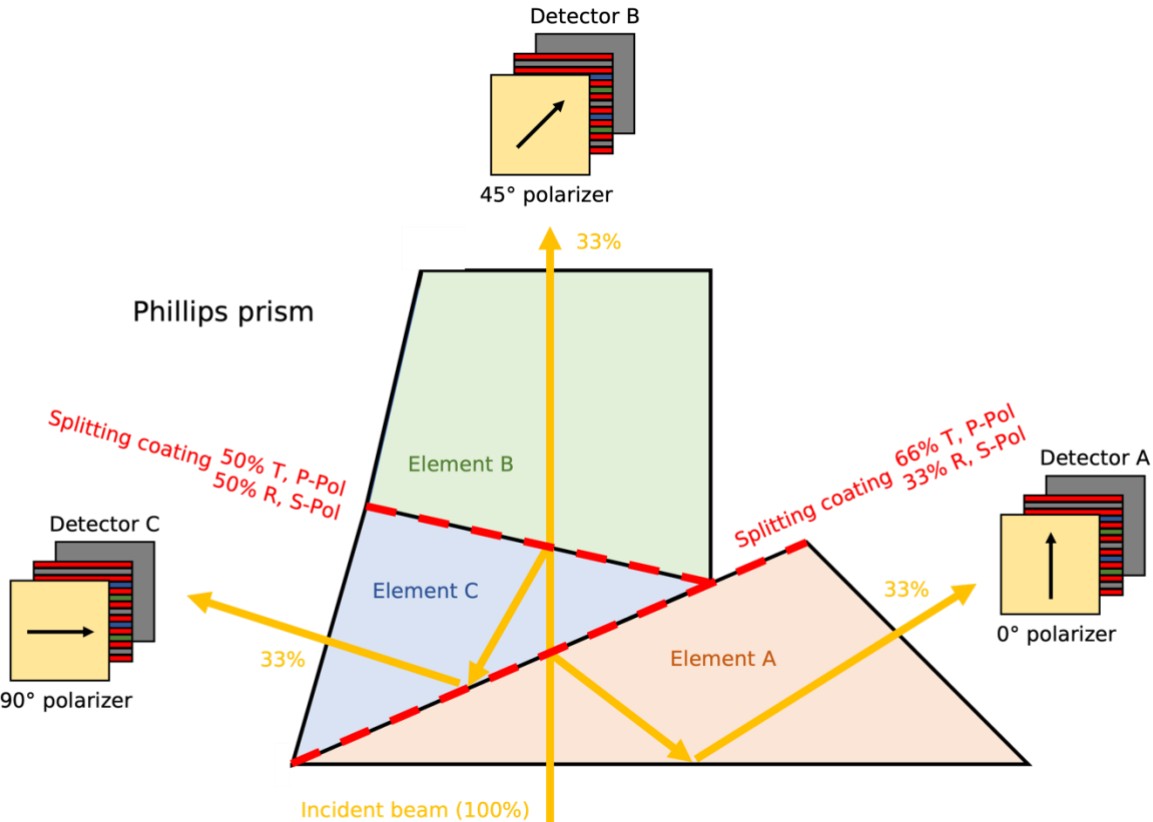

**Figure 2.** The Phillips prism is made of three elements: A, B, and C. Two splitting coatings split polarization states by transmission (T) and reflection (R). The coatings ensure that each HARP detector sees ~33% of the incident beam. The angle of the detector polarizer boosts the polarization efficiency of the prism along that light path. The light encounters the polarizer, stripe filter, then the detector FPA.




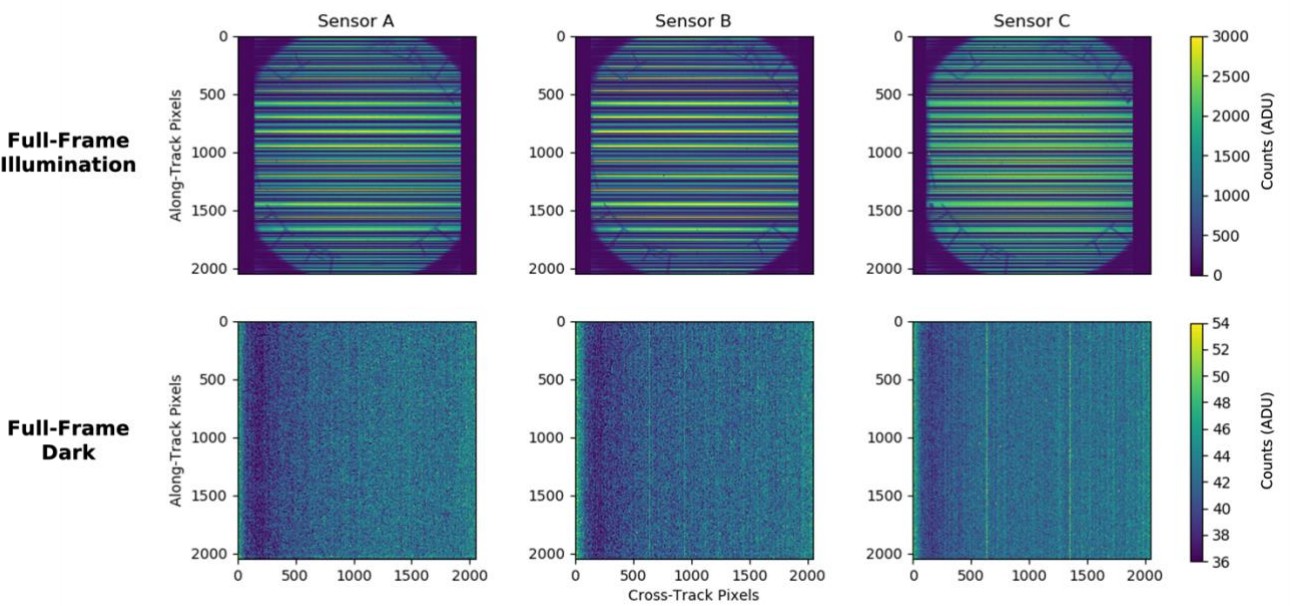

**Figure 3.** AirHARP captures a full-field raw image in each detector of the aperture of the NASA GSFC "Grande" integrating sphere (a) and a dark capture with the lens cap on (b). The dark shown here can be normalized and used as a template for any live data capture.



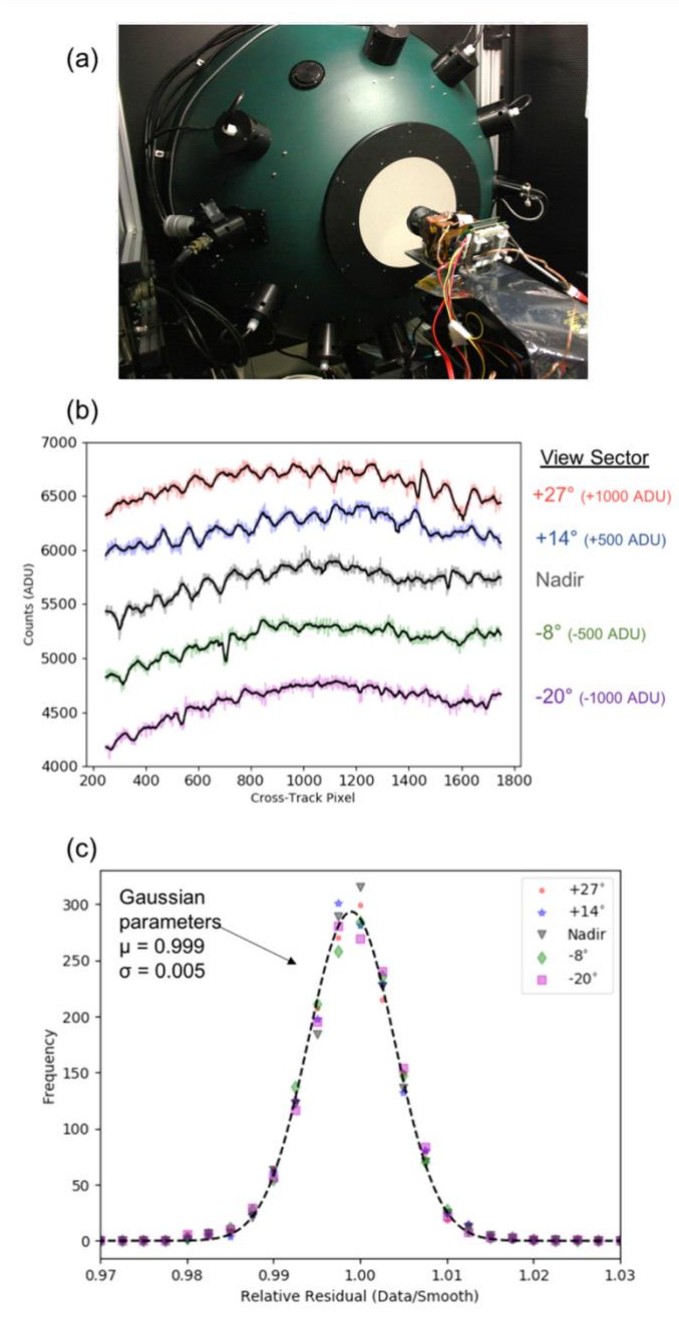

**Figure 4.** The flatfield is performed by submerging the wide field front lens into the aperture of an integrating sphere (a). This creates a full-field image similar to Figure 3a. In (b), the cross-track signal for several detector rows (colored data) is smoothed (black curves). After Eq. (3) is applied, only the SNR remains, which is normally distributed within 0.5% across the FOV. Data in (b) and (c) shown for AirHARP 670nm.




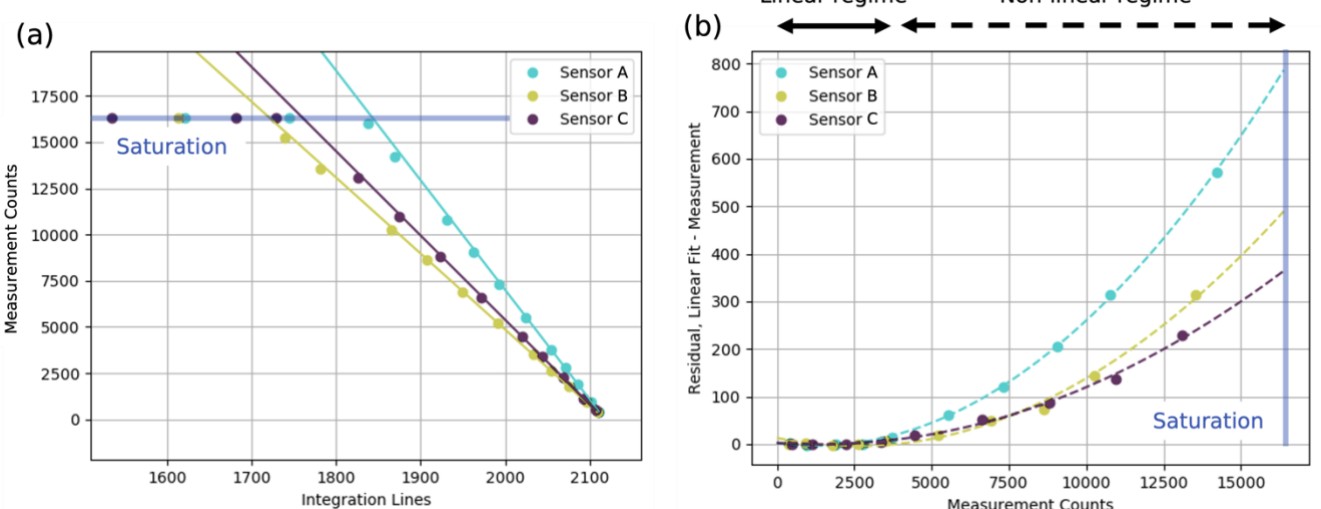

**Figure 5.** AirHARP detector integration time (compatible with "lines") is varied while imaging a stable light source (670nm channel shown). The counts in the linear regime (<3000 ADU) are fit in (a) for all sensors. This linear fit is compared to the entire dataset, and the residual is fit to a three parameter quadratic (b), which can now correct any raw measurement > 3000 ADU.



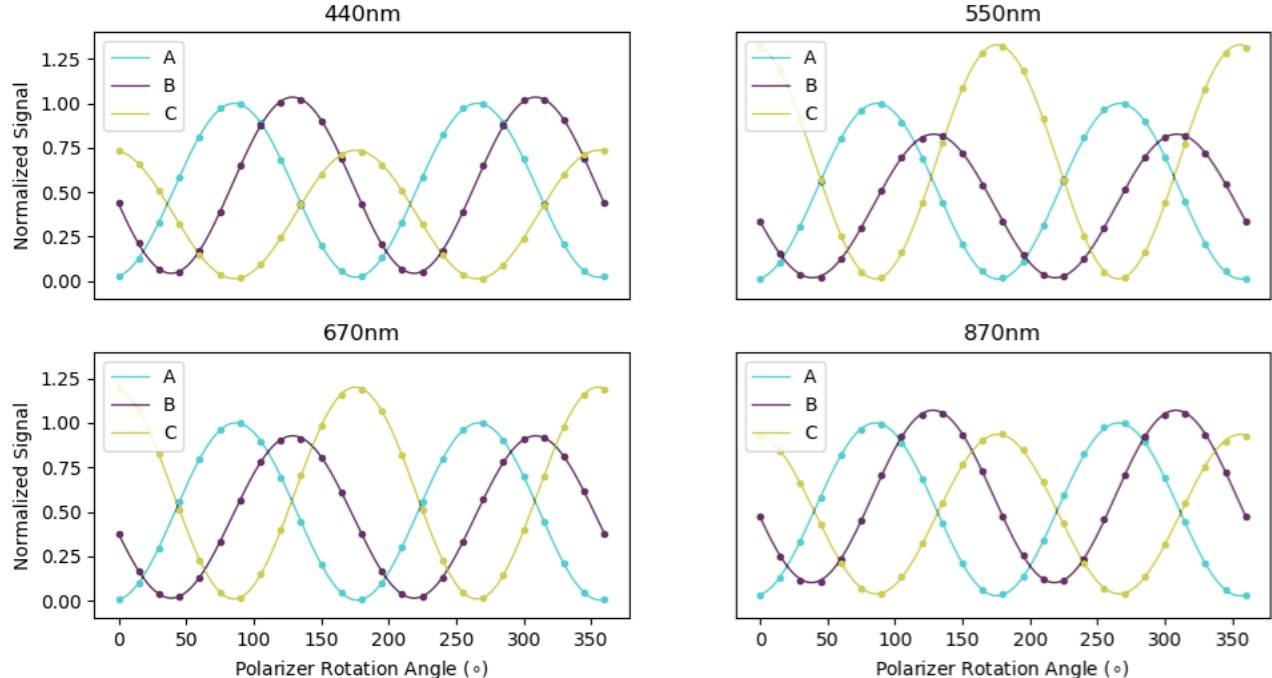

**Figure 6.** Malus curves for each of the four AirHARP channels. Each plot corresponds to an AirHARP channel, with data from Sensors A (cyan), B (purple), and C (yellow) fit to Eq. (12). These curves are normalized to the Sensor A maximum and represent the closest 4x4 nadir pixel bin, in each channel, to the AirHARP optical axis. Note the polarizer rotation angle is offset by -90° as shown.



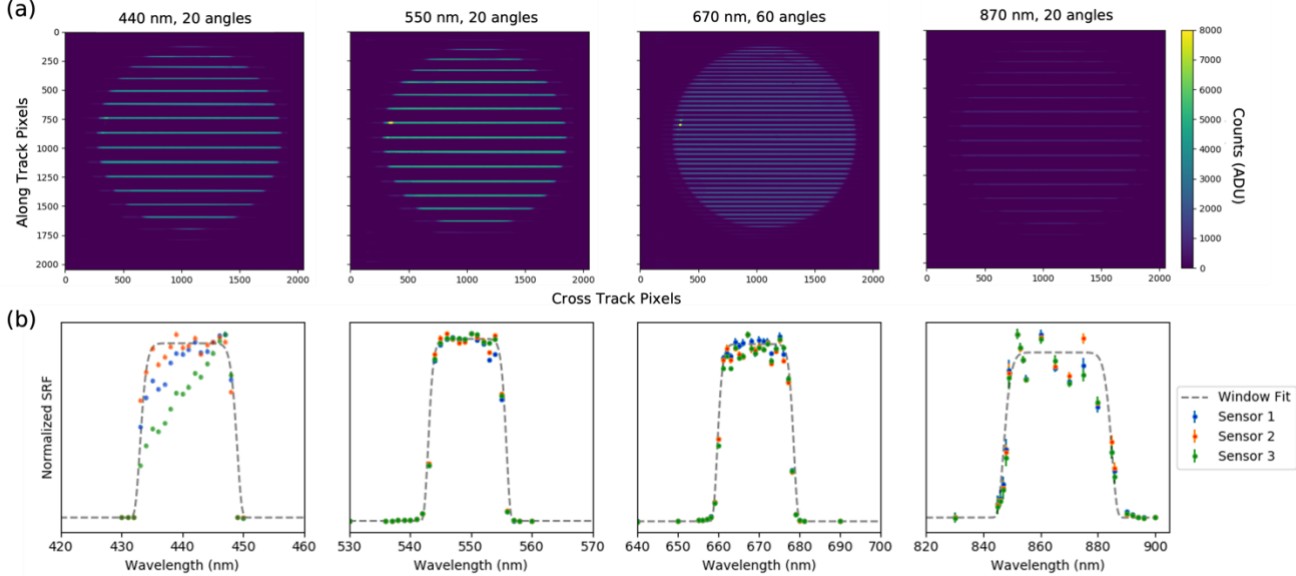


**Figure 7.** Examples of AirHARP images taken at different in-band Ekpsla wavelengths to show the distribution of illuminated stripes (a). The Ekspla power was weakest in the near-infrared, as evidenced in the 870 nm example (a, right). The AirHARP SRF for the three sensors and the super-Gaussian SRF fit (gray) is shown in (b). The panels in (b) correspond to the panels in (a). All data shown in (b) is normalized to 1 for each channel and sensor

individually.



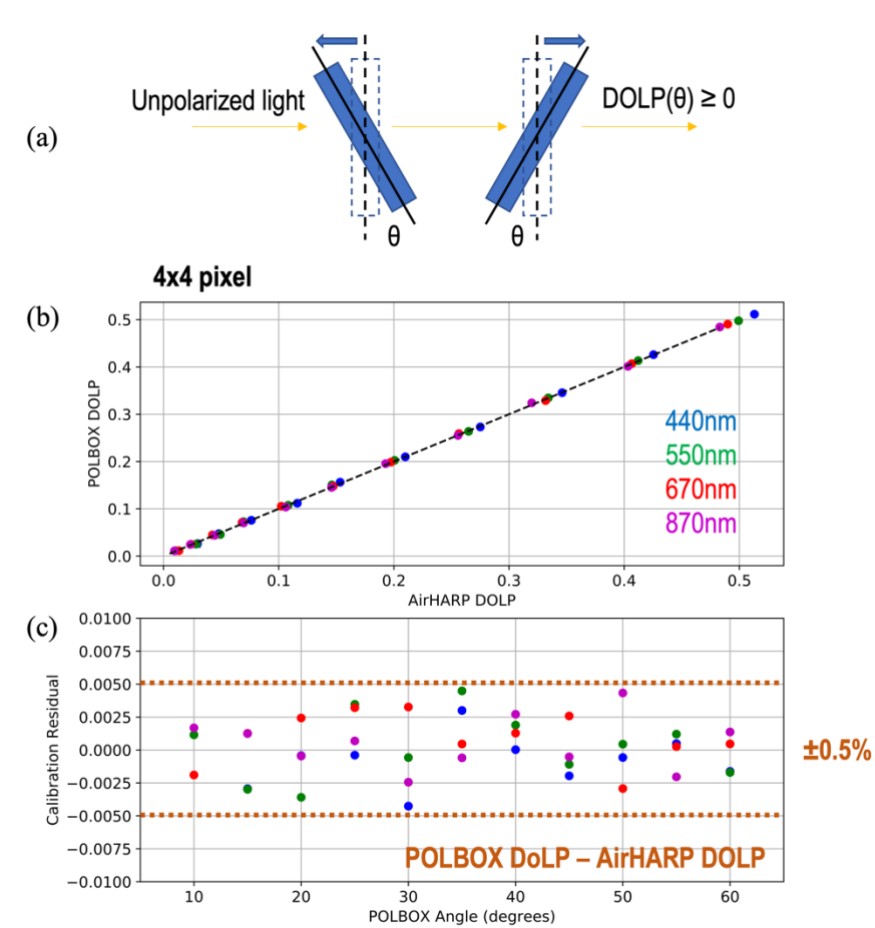

**Figure 8.** The POLBOX system generates partial polarization by rotating two glass blades (a). When comparing the DOLP theory to AirHARP measurement in all channels (b), we see AirHARP reproduces the entire POLBOX range within ± 0.5% DOLP (c). The lamp reflectance for this measurement was > 0.09 in all channels.







**Figure 9.** Multi-angle, co-located comparison between AirHARP and RSP for ACEPOL targets. Reflectance (left
column) and DOLP (right column) are compared for three compatible spectral channels: 550nm (top), 670nm
(middle), and 870nm (bottom). AirHARP data in the colors and RSP is black, with red signifying Rosamond Dry
Lake and blue data as ocean/glint cases. Error bars on the AirHARP data represent a 1-sigma standard deviation of
the superpixel bin.



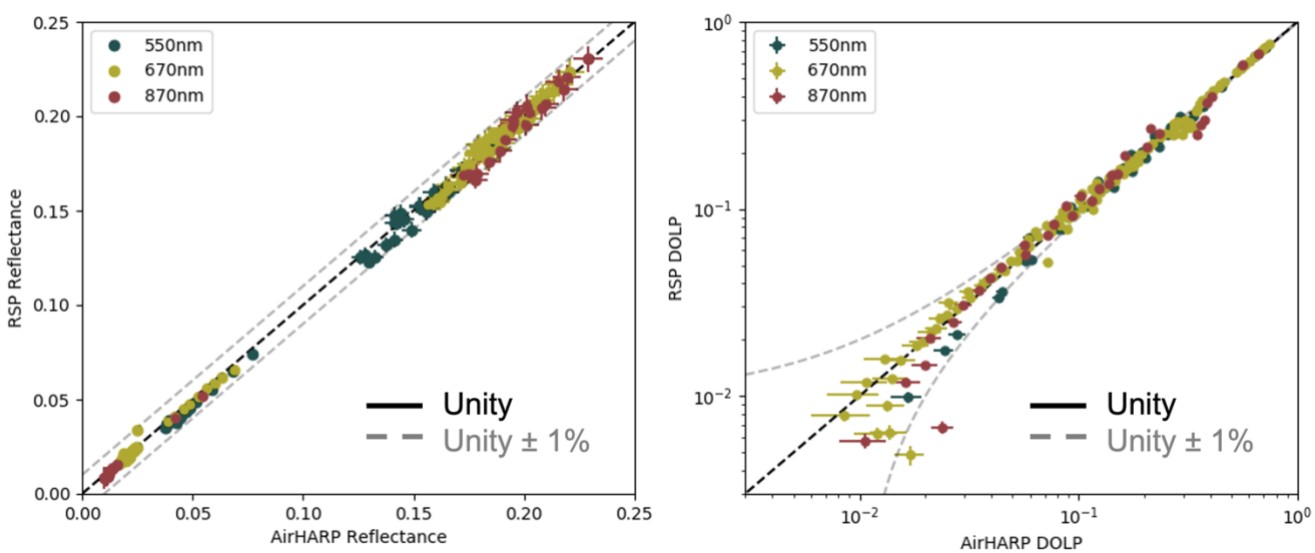

**Figure 10.** Direct comparison plot of AirHARP and RSP reflectance (left) and DOLP (log-scaled, right) for 550nm (green), 670nm (yellow), and 870nm (red) data over the two ocean and two desert ACEPOL cases shown in Figure 32. The dashed black line is unity, and the gray lines represent a 1% deviation off of unity. Errorbars represent uncertainty relative to the error models of both instruments explained in the text. All data shown represent co-located VZA up to 35°.
