# Peer review of "Pre-launch calibration and validation of the Airborne Hyper-Angular Rainbow Polarimeter (AirHARP) instrument"

_EGUsphere, 2023_

## Author Response (AR1)

**Author response to Anonymous Reviewer #1**

Reviewer #1 comments are in black, author comments are in blue, author revisions are in red.
* * *
I enjoyed very much reading the manuscript entitled "Pre-launch calibration and validation of the Airborne Hyper-Angular Rainbow Polarimeter (AirHARP) instrument". The contribution is well written, interesting and significant, and is scientifically sound. The subject is both timely and highly relevant for the community.

The authors thank reviewer #1 for the kind words, recommendation for publication, and remarks on the text and figures. We address their comments individually below and add additional remarks about where to find the revisions in text.

I have only two significant remarks for the authors to consider:

On page 7, it is mentioned: "In sensitivity studies on AirHARP dark image data, the dark counts do not depend on integration time, but are sensitive to operating temperature." This confuses me. The longer the integration time, the more dark counts should be registered, or what am I missing here? Or is meant that the dark counts per second does not depend on the integration time?

This is a good catch and as phrased, could be a little misleading. Dark counts are sensitive to integration time in general for CCDs. However, at the range of integration times we use operationally in flight (10-20ms), we do not see a significant variation in dark counts at a given temperature. In-flight detector temps were often on the colder side ( < 25 C, and in the lab we typically saw > 35 C), so the dark values were closer to the ADC bias level. During both LMOS and ACEPOL campaigns, we kept the instrument at a single 20ms integration time as well. We will change this sentence to discuss temperature only.

This discussion has been removed in the text. We felt that it would distract too much from the overall message, and this section isn't really about temperature or integration time. It is more about the dark subtraction itself, as a sub-algorithm.

In section 3.5.1, an Avantes spectrometer is used to correct the AirHARP measurements for any variation in Ekspla laser powerover the course of the testing period. Is it only used to monitor temporal variations of the laser power at a fixed wavelength, or also to check (and correct for) the difference in power between different wavelengths when scanning the Ekspla over one of the spectral bands of AirHARP? In the latter case, the spectral response of the Avantes itself should be taken into account. This could be explained in more detail.

The use of the Avantes in the SRF testing was the latter: to correct the AirHARP signal for the laser power of the integrating sphere at fixed Ekpsla wavelengths. The spectral response of the Avantes was accounted for prior to this test and correction. We hesitate to go into any further detail about the Avantes for brevity and focus, but this will be updated in the text.

On line 632 we added: "We set the Ekspla source at a given wavelength and verified each output channel and bandwidth using an external Avantes spectrometer. We use the spectrometer output to correct the AirHARP measurements for any variation in Ekspla laser power over the course of the testing period."

Apart from that, I only have minor, mostly textual remarks:

Some acronyms, while probably standard in the community, are not explicitly explained in the text, such as FPA, SRF, and VZA. For all clarity, they should we written out at least once when they are used for the first time.

The authors will go through the manuscript and address any acronyms that do not have a prior explicit reference.

We did a thorough find search through the text for these and other acronyms. Focal plane array (FPA) [line 179], spectral response function (SRF), [line 629] and view zenith angle (VZA) [line 857] have all been added at their first mention.

At the end of page 3 it is mentioned "...NASA PACE mission in 2023.", whereas earlier it is mentioned that the PACE mission will be launched in 2024.

This may be an artifact of when the paper was written versus submitted. The NASA PACE mission will launch in January 2024. This date throughout the paper will be harmonized.

We modified this to be: "A third, highly advanced version of the HARP concept, HARP2, recently launched on-board the NASA PACE mission (McBride et al. 2019). HARP2 achieves global coverage in two days and, as of this writing, has taken over 2 months of global radiance and polarization imagery. HARP2 anticipates a nominal mission lifetime of three years."

Section 2 on page 4: AirHARP has three detectors, not one. This is a bit unclear in the beginning of this section, where it is mentioned that the four channels are selected passively using a custom stripe filter on top of a charge-coupled device (CCD) detector FPA. It could be made clear from the beginning that AirHARP has three detectors each capturing a different angle of polarized light.

While it is mentioned later down the page and shown in Figure 2, I agree that an explicit mention of this in the first paragraph would be best. This will be updated in the text.

We modified this at this location to be: "AirHARP contains three CCD detectors that each image a component of the incident beam through the optical path. Each detector is covered by a linear polarizer, set at a unique angle."

At the top of page 20, a reference is made to Eq. 17, but I think Eq. 16 is meant: "It is convention to sometimes include an extra term in the denominator of Eq. (17) to account for the view zenith angle."

We have corrected this at this location to state: "We can divide Eq. (16) by the cosine of the solar zenith angle convert to TOA reflectance."

Labels (a) and (b) are missing in Fig. 3

These were added to Fig. 3 under the text lables on the left of the figure.

Figure 10: Unity is the black dashed line, but a black solid line is indicated in the legend.

Figure 10 is removed in lieu of a new analysis requested by Reviewer #2.

Minor typo's: Double space in section 1 on page 2 "improving our knowledge of microphysical properties", typo in Fig. 1 caption: "... which carries the same the 1.5U instrument", section 3.2 on page 8: "...will vignette photons toward the edge the FPA.", on page 9: "...where f is the valye of the flatfield correction", section 6 (Appendix): "in the above text defined as R and and DOLP is described below", strange page break in the middle of a sentence on page 32: "we found that the intercomparison with AirHARP did not...", on page 35: "Martjin Smit".

We thank reviewer #1 for these notes, and we will correct the text and figures to reflect them.

These have all been corrected in-line.

Caption of Fig. 4: it is not explicitly mentioned what is shown in panel (c).

The panel (c) shows the residual of the image counts after the normalized flatfield is applied to its generating dataset a la Eq. (3). This will be added in the caption with a identifier (c).

The identifier (c) was added to the figure caption.

**Author response to Anonymous Reviewer #2**

Reviewer #2 comments are in black, author comments are in blue, author revisions are in red.
* * *
This is a good manuscript that is within scope and of sufficient merit for publication after minor revisions. I appreciate the level of detail to which the authors describe the calibration and uncertainty assessment process for this class of instrument. Most of my comments address vagueness or sloppiness in the description with the intent of making the work more easily 'repeatable'. These should be easy to address.

The authors thank Reviewer #2 for their thorough pass through the manuscript and recommendation for publication. We address their individual remarks with comments below. If there are duplicated remarks, we will address the first and that will hold for all others.

I have two main complaints:

Figure 10b is used to demonstrate that AirHARP and the RSP instrument agree within 1% in DoLP. Because of the plotting log-scaling, I don't agree that one could conclude that from the figure for the majority of cases for which DoLP > 0.1. My comments below expand on this and offer suggestions on how to resolve.

I think the simplified AirHARP uncertainty model in the appendix is too simplified. Furthermore, the paper doesn't show how this model was derived based on previous equations. It also does not depend on scene reflectance, which it should. More details on this below.

These two points will be addressed in detail in later areas of this document.

These two points were addressed in the manuscript with a new analysis and error model. See below. We thank Reviewer #2 again for their kind suggestions and feel their comments led to a stronger paper.

I do think this is an important paper and commend the authors for clearly describing an approach that has been developed over several years. It will be an important tool as the PACE/HARP2 instrument and other similar polarimeters come online.

**Abstract:**

Line 18: clarify if RMS is defined in units of DoLP, or is relative. Also note RMS acronym should be spelled out. My personal preference is to use the terminology in Povey et al:

*Povey, A. C. and Grainger, R. G.: Known and unknown unknowns: uncertainty estimation in satellite remote sensing, Atmos. Meas. Tech., 8(11), 4699--4718 , https://doi.org/10.5194/amt-8-4699-2015, 2015.*

… in which case this statement would be more like "One sigma uncertainty of 0.25"

Line 19: Spell out FOV

The root-mean-square (RMS) uncertainty here is in units of DOLP. We will make both the acronym and units explicit in-text. Reviewer #1 also remarked on the acronym definitions, so we will address all in-text definitions and make sure they are first defined before they are abbreviated.

We changed this on line 16 to: "We show that this telecentric calibration technique yields a one sigma absolute uncertainty of 0.25% in degree of linear polarization (DOLP) in the lab for all channels and for pixels around the optical axis." The "one sigma" terminology is well-known in the science community, so we did not add the reference.

We also added the acronym to the earlier instance of FOV (line 12).

**Rest of the paper**

Page 2, line 44: again here you might want to point out that 0.5% is in units of DoLP, not relative as is the case for the radiometric calibration

We have revised this to: "0.5% absolute uncertainty in *degree of linear polarization"*

Page 2, lines 46-50, page 3 lines 55-60

Other SPEX references you might want to consider adding:

*Rietjens, J., Campo, J., Chanumolu, A., Smit, M., Nalla, R., Fernandez, C., Dingjan, J., van Amerongen, A., and Hasekamp, O.: Expected performance and error analysis for SPEXone, a multi-angle channeled spectropolarimeter for the NASA PACE mission. in: Polarization Science and Remote Sensing IX 34 -- 47) SPIE., 2019.*

*van Amerongen, A., Rietjens, J., Campo, J., Dogan, E., Dingjan, J., Nalla, R., Caron, J., and Hasekamp, O.: SPEXone: A compact multi-angle polarimeter. in: Proc. SPIE 11180, International Conference on Space Optics --- ICSO 2018 , 2018.*

Also note that the SPEX spectral spectral resolution is different for intensity and polarization. Polarimetric samples are every 10-40nm, depending on where this is in the spectrum.

Thank you for these citations and further information on SPEX. We will assess the papers and add them to the manuscript.

We have added the first reference as it is relevant. The second reference is redundant with Hasekamp et al. (2019), at least for the information on SPEXone.

Page 3, line 65: here you refer to the instrument MSPI, above it was described as AirMSPI

All instances of MSPI have been changed to AirMSPI [lines

Page 3, lines 71-72 AirMSPI needs to aggregate pixels to reach <0.005 DoLP. Further explored in

*van Harten, G., Diner, D. J., Daugherty, B. J. S., Rheingans, B. E., Bull, M. A., Seidel, F. C., Chipman, R. A., Cairns, B., Wasilewski, A. P., and Knobelspiesse, K. D.: Calibration and validation of Airborne Multiangle SpectroPolarimetric Imager (AirMSPI) polarization measurements, Appl. Optics, 57(16), 4499--4513 , https://doi.org/10.1364/AO.57.004499, 2018.*

*Knobelspiesse, K., Tan, Q., Bruegge, C., Cairns, B., Chowdhary, J., van Diedenhoven, B., Diner, D., Ferrare, R., van Harten, G., Jovanovic, V., Ottaviani, M., Redemann, J., Seidel, F., and Sinclair, K.: Intercomparison of airborne multi-angle polarimeter observations from the Polarimeter Definition Experiment, Appl. Optics, 58(3), 650--669 , https://doi.org/10.1364/AO.58.000650, 2019.*

Note the difference in conclusion of these two papers. It's not particularly relevant to this paper though.

Yes, we are aware of both excellent papers and have cited them elsewhere in the manuscript. We will add the citations here as well.

Both citations are added at line 89. Thanks again for the recommendation.

Page 4, line 95 I'm not familiar with the terminology "Alternative Vision". Can you provide a reference?

We apologize for any confusion here. Alternative Vision is the reference for this statement. It is a website with a technical article on the Phillips prism design. The author was not listed on the website. See citation below:

Alternative Vision: Separation Prism Technical Data, available at: https://web.archive.org/web/20070607142209/http://www.altvision.com/color_prisms_tech_data.htm, last access: 9 August 2021.

We decided to forego the reference, as it was a dead link. It supported a passing comment and was not particularly important for the rest of the discussion. Thanks for the catch.

Section 3.1: I like the differentiation between AirHARP, HARP2, HARPCubesat in the language and the use of 'HARP' as general description, but would have expected based on this that this section would use the language of HARP rather than AirHARP. Does that mean the detector

specification and background correction is different for the various instruments? Perhaps so, but I think it would have been nicer to include details on all three instruments in table 1.

** Thank you for this observation! The name juggling is a bit more systematic than we realized after submission. We will address any inconsistencies throughout the manuscript in the revision.

Reviewer #2 makes a keen observation that is important to the scope of the paper. The calibration theory is shared between all HARP instruments. However, the instruments and data can be different enough that going into too much detail could distract readers from the focus of this paper. For example, AirHARP and HARP CubeSat share a similar design, but have differences in performance, uncertainty, data compression, on-board storage, and data processing due to design choices and operational environment. HARP2 is a spiritual successor to HARP CubeSat. We consider it to be a different instrument entirely because many aspects of the HARP design were optimized.

Therefore, we decided to focus this work on AirHARP and reference HARP CubeSat or HARP2 at a top level only. Finally, AirHARP is the only HARP instrument with public data so far. By focusing on AirHARP only, this paper can be used to directly connect the calibration activities to the available AirHARP L1B reflectance products. HARP2 (and potentially HARP CubeSat) will see their own calibration and data papers starting next year. HARP2 was more extensively and rigorously tested than AirHARP as well, as it is part of a major space mission with desired performance goals.

We will refer to this response throughout this document as needed with the ** indicator.

We decided not to create a new table for the reasons above. HARP2 now has public data (PACE launched on February 8 2024 and data became public on April 11 2024), but it is still our position that AirHARP, HARP CubeSat, and HARP2 are different enough in performance and design that convolving them into a single paper would only confuse the reader or spread the analysis too thin.

A forthcoming paper by our colleague N. Sienkiewicz will go into further detail about HARP2 and the myriad of characterizations performed pre- and post-launch on that instrument. Even so, the calibration pipeline is still relatively the same for HARP2 as presented here for AirHARP – and that is the main aim of this paper – to introduce the overall methodology.

Figure 3: needs labels for (a) and (b) in the figure. Also there appears to be some sort of border irregularly surrounding the plots.

This figure will be modified to remove the irregular border and add these labels in the revision.

Figure 3 was revised for (a) and (b) row labels and to fix the transparency issues. Thanks for the catch.

Page 7, line 174 – AirHARP and HARP Cubesat have an internal shutter, does this mean PACE HARP doesn't?

See ** above. HARP2 does have an internal shutter, though it is different in several ways than AirHARP and HARP CubeSat. We will revise the language here to focus on AirHARP only.

We added some clarification to this line: "All HARP iterations have an internal shutter, which is actuated for in-flight dark captures."

Page 7, last paragraph: considering that the CCD dark current is different in the cross track direction, would have it made more sense to have oriented the CCD 90 from this so that the vignetted areas to use for a synthetic dark calculation are more uniform?

Not necessarily. We use a dark template from the lab, which is a full frame dark capture, as a basis for the synthetic correction. If we know the relative spatial distribution of the dark frame over the entire FPA, we could use any region of a vignetted detector to scale the template by temperature – as long as those two areas have a similar dark response. Therefore, there is no preferential orientation of the detector for the dark correction.

Page 8, line 198-199: "with a port fraction <5% and sphere multiplier >10 (PerkinElmer)" uses terminology that may not be familiar to all readers – please define port fraction and sphere multiplier.

We will add to this paragraph that the port fraction of an integrating sphere is the ratio between the internal surface area and the area of open holes and apertures. The sphere multiplier characterizes how much the reflective surface and port fraction of the sphere contributes to increasing the light output relative to the source. We will also add the PerkinElmer citation, which we accidentally omitted from the reference list.

We ultimately decided to remove this reference and statement in the text. Like the reference for the Phillips prism, it was supporting a passing comment that doesn't make or break the paper.

Page 8, line 206: I'm confused by what appears to be a link: andor.oxinist.com, which is actually a reference.

We apologize for the confusion. We will change this citation to "Oxford Instruments Andor". Like the Alternative Vision reference, it is a technical article with no lead author and no publication year given.

We also removed this reference for similar reasons above. We did not go into detail about optical etaloning in this work, for scope reasons.

Page 10, line 260: Wouldn't it have been simpler to just represent this as integration time instead of 'integration lines'? It is confusing at first glance because larger values mean opposite things. For example the statement "Sensor B saturates earlier than Sensors A or C" is confusing because the lowest 'integration line' corresponds to the longest integration time, if I understand correctly.

Yes, we will make this change in the revision. Integration lines represent the integer values we used in the HARP instrument firmware to set integration times for each detector, and in hindsight, they are not as intuitive as the actual integration times in a plot like this. Thanks again for the suggestion. And your understanding is correct – larger integration lines correspond to shorter integration times.

Thanks for this observation. We modified Figure 4a to show integration times vs. counts. It is more immediately intuitive to the reader. We changed all reference of "integration lines" to "integration times" starting at line 256 to say: "The integration times of each sensor are increased, and images are taken until all three sensors and channels saturate… …The standard process is used to form a template image at each integration time and for each detector. We take a small pixel bin (~4x4) along the optical axis in the templates and plot those values against their integration times… …There is a monotonic, positive relationship between integration time and detector counts, up until the saturation point, $2^{14}$ ADU."

Section 3.3: If possible, add a sentence or two explaining the likelihood that doubling the integration time is identical to doubling the sphere intensity. This test, which varies exposure time but not intensity, relies on this.

Yes, this is true and a core tenet of the non-linearity part of the pipeline. We will add this to the text is some fashion. Beyond this, the non-linear correction allows us to modify the detector integration time on the fly during a field campaign or space mission, with the knowledge that the detector response will be predictable.

We added this at line 278: "Non-linear correction early in the calibration pipeline allows for a verification of the reciprocal test during absolute radiometric calibration (counts measured at a single integration time across a variety of lamp levels)."

Page 11, line 264: how is the 'linear region' defined at 3000 ADU? I think I understand what you are doing in Figure 5b that expresses this, but feel like you need another sentence or two to explain how you're going from fig 5a to fig 5b

The linear region upper bound at 3000 ADU is the location where the fit residuals are no longer randomly distributed around zero. The bound itself depends on the resolution of the data and is empirically defined. We will add more detail on this in the text.

We added this at line 263: "We identify a set of data points with minimal deviation from a linear response (<3000 ADU), and compare the linear fit over those points to the rest of the data."

Equation 4: I was confused by this equation until I realized that the DNflat on the left hand side of the equation is different from that on the right hand side. The left hand side is the value of DNflat from the linear *fit* to the data, not the actual data. If my understanding is correct, I suggest you use some notation to indicate that they are different things.

We apologize for the confusion here. The entire left side of the Eq. 4 is the fit in Figure 5b. DN_corr is the linear fit from Fig 5a and DN_flat is the counts data from Fig 5a. We will separate the left side into a separate equation DN_fit = DN_corr – DN_flat, and have the current Eq.(4) be DN_fit = the polynomial function.

We added some notation to simplify the Equation: "

$$DN_{corr} = DN_{linfit} - DN_{BC} = n_0\,DN_{BC}^2 + n_1 DN_{BC} + n_2, \qquad (3)$$

where $DN_{corr}$ is the non-linear corrected counts, $DN_{linfit}$ is the fit performed on the linear region, $DN_{BC}$ is the counts data derived from Eq. (1), and fit parameters $n_0$, $n_1$, and $n_2$ are free parameters."

Page 16, line 396: Can you clarify if the polarimetric coefficients are determined for all pixel bins, or just the nadir bin?

The polarimetric coefficients shown in this work are determined for the nadir bin, and we leverage the telecentric optical train/flatfield to spread those coefficients to other FOVs. We will clarify this in the text of this section.

We added text to line 385: "The **C** translates normalized, corrected detector counts to normalized Stokes parameters for pixels along the optical axis (though applicable to polarization measurements anywhere in the FOV)."

Page 16, line 398: do you mean 'deviate' rather than 'derivate'?

Yes, this was changed in line to 'deviate'.

Page 17, line 412: you're referencing equation 5, but do you mean equation 10 instead?

Yes, this change has been made in text. However, Eq. (14) could be used in general for any co-located target between the three detectors.

Page 17 line 413: is sigma_c_ij really from Table 2a rather than 2b?

Yes, this is a typo. Sigma_C_ij refers to the Table 2a. We will address this in the revision.

This was changed to Table 2a on line 410.

Page 18, line 440: I suspect that you're referring to a different equation here than eq 5, suggest you verify all references to equations are correct

We will address the above comments to Page 16, 17, and 18 in the revision.

We did a thorough find-search through the text to harmonize in-text references to equations.

Page 18, line 442: I thought you were referring to "Detector A, B, C" rather than "Sensor 1, 2, 3". Is there a difference? If so, explain, if not, please make the terminology consistent

Detector [A, B, C] and Sensor [1, 2, 3] mean different things. Detector [A, B, C] refer to the physical FPAs behind the each port of the prism. Sensor [1, 2, 3] refers to the light path through the instrument that corresponds to Detector [A, B, C]. This is defined in Section 2, line 108. It is a nuanced definition and we will address any inconsistencies in the text in the revision.

In this section, the SRF is calculated at the Sensor level because there is a possibility that the SRF in Sensor 1 is not the same as Sensor 2 or 3 (as we see in the 440nm). The actual Detectors play a role in the SRF, but only from the perspective of quantum efficiency (shown in Table 1). The system SRF is a combination of the detector QE, stripe filter, prism response, and multi-bandpass filter.

Page 20, line 484: "the illumination of the nine lamps is linear" – does this mean that each lamp adds linearly to the total illumination?

Yes, and we will add a statement about this in the revision.

We added this statement to the text: "The spectral sensitivity of "Grande" peaks around 1 μm, and each of the nine lamps add linearly to the total illumination."

Page 21 line 501 and equation 17: If gamma doesn't mean the same thing as in equation 11, I suggest using a different parameter. If it does, please note this.

Great catch, these are different parameters. We will use an epsilon in Eq. 17 instead.

The equation now has an epsilon for the linear bias.

Page 21 line 504 "compatible with zero within 3 sigma" what does this mean? Does it mean 'equivalent to zero'? I presume so, but I don't know what within 3 sigma means in this context.

We do not show this plot in the paper, but the radiometric calibration is a linear fit between the calibrated radiances of the NIST-tracable integrating sphere and the relative intensity (Eq. 17 terms in parentheses) measurements of the sphere light by AirHARP. The calculated uncertainty

on each fit parameter (k and gamma) give an estimation of how well the parameter describes the data. The gamma term is equivalent to zero within 3 times its fit uncertainty. We will remove the "sigma" definition and explicitly describe this in the revision.

We added to this section: "For all channels, the linear bias $\epsilon$ is compatible with zero within three standard deviations of the least-squares fit error on this coefficient."

Page 23, line 560: references characterization in "late 2022" Is this a typo, or if this already happened please rephrase (and note where the error model is documented)

Thanks for the catch here, and this shows how long this paper was in development! We will remove this sentence altogether for reasons in ** above. However, this test was done for HARP2 and researchers in our group are currently assessing the data. The results for ** reasons are beyond the scope of this AirHARP work. The AirHARP error model has not been documented publicly. We will address the reviewer #2 concerns on the error model in later sections below.

This sentence was removed. For information on the error model, see below.

Page 24, lines 571-9: I think you reference them earlier, but this would be a good place to put references for the LMOS and ACEPOL field campaigns as well.

Yes, this makes sense. We will add these citations here in the revision.

We added the ACEPOL (Knobelspiesse et al. 2020) and LMOS reference to the paper:

Stanier, C. O., Pierce, R. B., Abdi-Oskouei, M., Adelman, Z. E., Al-Saadi, J., Alwe, H. D., Bertram, T. H., Carmichael, G. R., Christiansen, M. B., Cleary, P. A., Czarnetzki, A. C., Dickens, A. F., Fuoco, M. A., Hughes, D. D., Hupy, J. P., Janz, S. J., Judd, L. M., Kenski, D., Kowalewski, M. G., Long, R. W., Millet, D. B., Novak, G., Roozitalab, B., Shaw, S. L., Stone, E. A., Szykman, J., Valin, L., Vermeuel, M., Wagner, T. J., Whitehill, A. R., & Williams, D. J. (2021). Overview of the Lake Michigan Ozone Study 2017. Bulletin of the American Meteorological Society, 102, 12, E2207-E2225. https://doi.org/10.1175/BAMS-D-20-0061.1, 2021

Page 24, line 591: does this require a priori information on wind speed / surface roughness?

Not for the cross-comparison shown in this work. We know the target is sunglint from visual context in the AirHARP data and the geometry of the observation, but we are only interested in the closeness of the AirHARP-RSP angular matching over the same targets.

Page 25, line 600: perhaps some citations for the RSP instrument here, specifically about calibration?

We will add van Harten et al. (2018), Knobelspiesse et al. (2019), Smit et al. (2019), and Cairns et al. (1999) citations, which are a good spread of calibration work using RSP over the years.

The first three citations were added, the last was cited when RSP was introduced.

Page 25, line 610: is the RSP footprint a circle, or an oval? There is the IFOV, but an integration time that makes it like an oval. The Knobelspiesse 2019 paper you cite has a model for this as well as McCorkel et al 2016. It would potentially make your fits slightly better.

McCorkel, J., Cairns, B., and Wasilewski, A.: Imager-to-radiometer in-flight cross calibration: RSP radiometric comparison with airborne and satellite sensors, Atmos. Meas. Tech., 9(3), 955--962 , https://doi.org/10.5194/amt-9-955-2016, 2016.

The true RSP footprint is an oval due to bi-directional smear, as discussed in both papers. An important thing to note though is both papers are intercomparison papers first. Our paper only uses the AirHARP/RSP intercomparison as a top-level demonstration of our nadir calibration spread to all FOVs. Therefore, we simplified the intercomparison here in several ways. One of those simplifications is the RSP footprint as a superpixel square. This said, converting this to an oval as in Knobelspiesse et al. (2019) and McCorkel et al. (2016) is a reasonable suggestion by reviewer #2, and would not require a major revision. This may improve the Figures 9 and 10 in lieu of their later comments and has basis in the literature.

The revised intercomparison technique uses a smear oval like Knobelspiesse et al. (2019). We felt it important to preserve the native footprint of the RSP from ER-2 altitude (220m) relative to AirHARP (7x7 circle, each pixel is 20 m ground res. in L1B) and then linearly decrease the weighting of the pixels in the heading direction (forward and aft). The smear is 110m in both directions from the edge of the main core. The following is an example of the smear oval mask used in the intercomparison:

[Figure]

The AirHARP-RSP pixel match lies at the center of the oval. This mask is multiplied directly into the AirHARP image at the location of the pixel match once a suitable match pixel is found. The "superpixel" standard deviation is mask-weighted as well. This mask is applied to geometry (VZA, SZA, VAA, SAA) and geophysical parameters (I, Q, U, DOLP).

Page 27, line 663: only Fig 10b appears to be log-scaled

This was done to emphasize the comparison at the low end of the DOLP, which is part of the focus of this paper. Polarimetric uncertainty and intercomparisons in this DOLP region are of considerable interest to the climate community, as modern aerosol and ocean color property retrievals may rely on accurate measurements at low DOLP.

Figure 10 has changed substantially. See discussion below.

Figure 10 refers to "Figure 32" which isn't a part of the paper. I presume you mean Fig 9

Yes, we will correct this in the revision.

This is corrected.

Figure 10 (right): This figure doesn't convince me that RSP and AirHARP agree within 1% for the log scaled DoLP. Because of overplotting, it is impossible to tell if the difference is beyond 1% for DoLP > 0.1 (possibly majority of observations based on Fig 9). It would have been better to plot the difference between RSP and AirHARP. It would have been even better to consider how that difference compares to the paired RSP and AirHARP uncertainty estimates. Validation of the uncertainty model is what is most important for algorithms which will use that model in retrieval. The models are not fixed: for RSP, the uncertainty varies with DoLP and reflectance, for AirHARP the simplified model shows it varies with DoLP (see later comment about the AirHARP simplified model). Furthermore, you cite the papers (Knobelspiesse et al. 2019, Smit et al. 2019, van Harten et al. 2018) that I'm not sure support the assertion that the difference is 'reasonable' since they relate to other instruments besides AirHARP. However, they do provide examples of more in depth analysis on this.

So, my recommendation is to redo figure 10b to show the difference between RSP and AirHARP scaled by the squared sum of the uncertainty estimate for the given DoLP and Reflectance value. For one sigma uncertainties 67% should agree within those bounds. You could also check with 2 sigmas / 95%.

A 1:1 comparison was chosen for this paper instead of a direct difference because of the paper scope. We wanted to emphasize the overall comparison and not the angle-to-angle differences that are front-and-center in a direct difference plot, especially with only four data cases.

However, revisiting this analysis is a reasonable request from reviewer #2. If we were to show a difference plot in this paper as suggested, we would prefer to include more ACEPOL datasets

than the four cases in the current version. We suspect this is a larger revision. It would avoid overemphasizing any granule-specific or angle-to-angle differences in the intercomparison though. Because reviewer #2 only suggested minor revisions, we will weigh the effort vs. benefit of redoing this part of the paper and perform any reasonable re-analysis.

If it is found that this amounts to a larger revision, we still agree that the interpretation of Figure 10 as noted by Reviewer #2 could be stronger. We will assess the readability, clarity, and overplotting effect of these plots. One solution in this direction could be to separate the plots by wavelength (as well as re-analyze with the oval RSP footprint and uncertainty model below). This would reduce the impact of point crowding.

Thank you for these suggestions.

We decided to use seven granules to balance minor revisions with the need to be more quantitative with the intercomparison. 2 ocean, 5 desert (which includes the entire rosette over Rosamond.) We also changed the intercomparison from a search to find the best matching measurement and a cost function minimization to a direct angle-to-angle match using the smear oval. This allowed for the error-normalized difference study as requested by Reviewer #2. This change was necessary because it would not make sense to do an error-normalized difference on data where we explicitly looked for matches in I and DOLP. The changes are too numerous to copy and paste here, and we invite Reviewer #2 to look into the revised text for details.

Now, the differences are tied to the details of each instrument's error model and how much we trust the geometry and georegistration. It is a bit more faithful to a true intercomparison. The results in Figure 10 are more robust statistically. However, there are a lot of potential error sources which are convolved or blurred over in this analysis. Either way, thank you for the suggestion. I believe this change made for a stronger paper.

Appendix A, line 760

You show a 'simplified' AirHARP error model which is what an algorithm developer would use. Presumably this comes from equation 19 and 14, but those are not expressed in terms of reflectance or DoLP. This appendix should show the full model for reflectance and dolp, and then the simplified model as you show.

Additionally, I am concerned the simplified model you show is too simplified. The DoLP uncertainty does not depend on shot noise or reflectance, and I find that that drives performance for scenes over the ocean. The (Knobelspiesse et al. 2019, Smit et al. 2019, van Harten et al. 2018) papers you cite all focus to some extent on this. Of course, the binning of AirHARP compared to RSP will drive down shot noise. Random uncertainty is reduced by sqrt(n), so I am also confused why you have the 1/B term in equations 26 and 27 rather than 1/sqrt(B) or similar.

Finally, I would have expected some decreasing performance from the center for where the calibration is performed to other pixels for which it is extended. Is this expressed in the calibration coefficients? Knowledge on this characteristic if it is indeed different would also be useful for algorithm developers who can account for this in retrieval algorithms.

Since the submission of this paper in May 2023, we have advanced the error model for AirHARP, though it is still currently unpublished. Adding a more involved error model based on the earlier equations is also a reasonable request from reviewer #2, given the utility in retrieval studies and public availability of AirHARP data. This will be adjusted in the appendix in the revision, and address reviewer #2's other concerns here.

As far as off-nadir performance goes, we can see from Figure 4c that the flatfield "flats" its generating dataset across the FOV within 0.5%. This distribution is comprised of random noise. The flatfield and telecentric physics are the vehicle that allows the nadir calibration spread to other FOVs. Therefore, our current lab analysis suggests that there is negligible off-nadir degradation in response. This may change in the field data due to interpolation during data processing and variable FOV effects like lens condensation or pitch surfing mentioned in Section 4. These impacts have not been fully characterized as of this work. Further studies are underway on these effects. Section 4 intercomparisons demonstrate the off-nadir performance in the field from a data perspective, which also suggests the performance is comparable at nadir up to 35 degrees in VZA. Adding more ACEPOL datasets to the comparisons in Figure 10 may improve the lab and field assessments of off-nadir performance as well.

Forthcoming papers will go into detail about off-nadir performance from a laboratory perspective for HARP2. Similar studies were not done at the same level for AirHARP for time or resource reasons.

We also added more detail to the error model in the Appendix, with derivations and descriptions from earlier equations. A full error model is out of scope for the paper, and it would amount to a major, major revision to add one at this point. Also, it is not a simple task for a data user to go from the L1B dataset to detector counts and uncertainties, so we decided to develop an approximation based on the L1B products themselves. This L1B error model performs quite well in our error-normalized difference study. It is again too much to copy/paste here, but I invite you to look at the revisions in detail.

Author response to D. J. Diner

D. J. Diner's comments are in black, author comments are in blue, and author revisions in red.
* * *
This is a generally well-written paper explaining the AirHARP calibration process. Several detailed comments/questions are below.

The authors thank D. J. Diner for their time and thoughtful review of the manuscript. We will respond to each comment individually and in the case of small edit recommendations, we may chunk several comments together and respond underneath.

Lines 65-72: For consistency with earlier text, suggest changing "MSPI" to "AirMSPI".

Line 67: Change "gimble" to "gimbal".

Line 68: Change "FPA" to "focal plane array (FPA)" (spell out acronyms when first used).

Line 68: The AirMSPI FPA is two-dimensional but not used in the sense of a framing area array. Perhaps change "two-dimensional" to "pushbroom".

We will address these minor errors in the manuscript in the revision (and any others mentioned below). Thanks for the recommendations.

These were corrected in text.

Line 81: I believe PACE launch is now in 2024.

Yes, the PACE launch is currently set for January 2024. We will update this reference.

We modified this to be: "A third, highly advanced version of the HARP concept, HARP2, recently launched on-board the NASA PACE mission (McBride et al. 2019). HARP2 achieves global coverage in two days and, as of this writing, has taken over 2 months of global radiance and polarization imagery. HARP2 anticipates a nominal mission lifetime of three years."

Lines 95 and 832-833: Check the Alternative Vision link. I couldn't get it to work.

One of the reviewers also mentioned the link was dead. We will refer to Borda et al. (2009) instead in the revision.

We decided to forego this reference.

Line 150: The Semiconductor Components Industries 2015 reference is missing.

Thanks for catching this, we may have omitted this by accident. The reference is a technical spec sheet and is here: https://www.imperx.com/wp-content/uploads/2017/08/KAI-04070-D.pdf

We will add this to the references in the edit.

This reference was added.

Line 172-173. If the dark counts are sensitive to operating temperature, then they are either related to dark current, or there is some offset in the video signal that is temperature sensitive. Counts attributed to dark current would presumably be sensitive to integration time. Do you have an explanation of why the dark counts are not integration-time dependent?

Copied from the response to Reviewer #1, who also commented on this –

This is a good catch and as phrased, could be a little misleading. Dark counts are sensitive to integration time in general for CCDs. However, at the range of integration times we use operationally in flight (10-20ms), we do not see a significant variation in dark counts at a given temperature. In-flight detector temps were often on the colder side (< 25 C, and in the lab we typically saw > 35 C), so the dark values were closer to the ADC bias level. During both LMOS and ACEPOL campaigns, we kept the instrument at a single 20ms integration time as well. We will change this sentence to discuss temperature only.

Equation (2) and surrounding text: It might be helpful to put a subscript on the delta parameter to show that it is pixel dependent. Does the $DN_{raw\,[0-200,1848-2048]}$ represent a mean (or perhaps median) of dark values over pixels 0-200 and 1848-2048? Are 0-200 and 1848-2048 the full range of masked pixels, because in line 209 you state that the active science area is pixels 200-1800.

We can see how this notation could be somewhat confusing. We propose this new format for Eq. (2):

$$DN_{dark} = \widehat{DN_{dark,lab}}\ \delta$$

This is a better notation because the dark lab template (with the carat to signify normalization) is the workhorse of the correction. The delta would be better served as a factor that comes from the vignetted area of the sensor in the respective live data capture (and further explained in text).

The 0-200 and 1848-2048 are the active science pixels cross-track. The 1800 on line 209 is a typo, we will change in the revision. Thanks for catching that.

We added this discussion to the text: If we cannot take dark captures on-orbit or during field campaigns for any reason, we can create a *synthetic dark* by scaling a normalized dark template from the lab by an average of all along-track counts in the vignetted areas of a live data capture (typically over cross-track pixel indices 0-100, seen in Figure 3b):

$$DN_{dark} = \widehat{DN}_{dark}\overline{DN}^*_{raw} \tag{2}$$

where $DN_{dark}$ is the estimated dark image counts and $\widehat{DN}_{dark}$ is a normalized dark template image from the lab. $\overline{DN}^*_{raw}$ represents a spatial mean of pixels in the vignetted area of a raw image capture (similar to Figure 3a)

Lines 198-199: Define the terms port fraction and sphere multiplier. The Perkin Elmer reference is missing.

Copied from the response to Reviewer #2, who also commented on this –

We will add to this paragraph that the port fraction of an integrating sphere is the ratio between the internal surface area and the area of open holes and apertures. The sphere multiplier characterizes how much the reflective surface and port fraction of the sphere contributes to increasing the light output relative to the source. We will also add the PerkinElmer citation, which we accidentally omitted from the reference list.

We decided to forego this discussion and reference for scope/brevity.

Line 204: The smoothing process could cause individual pixels with greater or less responsivity to have a residual calibration artifact. Has this been considered?

This has been considered in the flatfielding process, but as of this work, only the pixels with large offsets relative to the smoothed response (typically defects/contaminations, etc.) are addressed. These pixels are masked as the imagery moves through the L1B pipeline. A more sophisticated quality flagging will be in place for HARP2 calibration/L1B products next year.

That said, most of the pixel-to-pixel relative structure is described by the smoothing window in Figure 4b. In general, these statistics across the FOV are distributed within 0.5% (1 standard deviation of the Gaussian distribution of residuals after the flatfield is applied back to the generating dataset). Binning pixels in L2 retrievals, as was done in Puthukkudy et al. (2020), McBride et al. (2020) will further smooth the variation in sensitivity between neighboring pixels.

Line 206: The andor.oxinist.com link does not work.

We will find a new reference. Thank you for catching this.

We decided to forego this reference.

Line 209 and Figure 4b: If the active area is pixels 200-1800, why don't the plots cover this full range? Is this because of the sliding smoothing window?

Yes, the smoothing window algorithm used here may create artifacts at the edge of the active science area, since the vignetted portion of the sensor contains far lower counts. One way to solve the edge artifacts is to mask the vignetted area and linearly extrapolate the flatfield at the edges of the science area by using the trend of the last 10 or 20 pixels in each direction. We will revisit these plots in the revision.

Line 218 et seq.: Perhaps put subscripts on the flatfield correction factor $f$ to indicate that it is pixel (and row) dependent.

All terms in Eq. (3) are pixel-dependent and this is explained in text.

We modified this text: "The optical axis is chosen specifically as the location of $f$=1 to simplify the later steps in the calibration process that also use optical axis pixels. We then apply the flatfield correction at the pixel-level:

$$DN^* = \frac{DN_{corr}}{f(x,y)}, \tag{4}$$

where f is the value of the flatfield correction for that pixel, which are a function of along-track and cross-track pixel indices x and y, and the numerator of Eq. (4) is the same as Eq. (3)."

Line 234: Are the optics perfectly telecentric, so that the AOI on the detector is exactly 0º for all pixels? In practice there is generally some slight non-telecentricity on the order of a fraction of a degree to a few degrees.

The optics are telecentric to less than 1° over the entire image plane, with a large majority of chief rays within 0.5°.

Section 3.3: The linearization curves are derived by changing integration time, but not illumination level. Have you done any tests to verify that reciprocity holds? Why is linearization done after flat-fielding correction; shouldn't linearization occur first?

Yes, we have analyzed sphere radiance data from AirHARP without the non-linear correction applied and we see a similar behavior when varying the sphere illumination level at a constant integration time. The reason why we vary integration time in this test is to verify that we can change integration time on orbit and expect a linear gain response (after this correction is applied).

This could be done the other way around, but it requires more degrees of freedom in the test environment. Light source illumination needs to be cycled for each integration time, and each new light level requires time to stabilize. Therefore, it is not as efficient as varying integration time for a stable illumination level.

And yes, since linearization is relative to the ADC, it should be done before any superficial corrections, like flatfield or polarization calibration. Here, we are studying linearization along the AirHARP optical axis where the flatfield is set to 1, so it is effectively the same thing.

Line 418: Define what a superpixel is, as this is the first time this term is introduced.

We added this to the text: "At the integration times we use, shot noise and potentially scene spatial variability can dominate, so the standard deviation of data from a real AirHARP superpixel, a rectangular, connected set of along-track and cross-track pixels, is used."

Line 424: Spell out SRF when first used (applies to all acronyms).

We will do a dedicated pass through the manuscript and define all hanging acronyms. The other reviewers also emphasized this revision.

This and other acroynms were explained and harmonized with their first use.

Lines 444-446. Text indicates that narrowing of the leading edge of the 870 nm channel was discussed earlier, however I could not find such discussion.

Thanks for the catch. This may have been omitted from an earlier section before submission but the reference remained. We will revise this in the edit.

We removed this reference from the text.

Lines 446-545. The rolloff at the short wavelengths of the 440 nm band changes the effective band center and bandwidths of this channel. I don't follow the logic "Rayleigh-like" SRF adjustment. Why not simply represent this channel by its effective spectral parameters that roughly represent the same radiometric output as the actual SRF?

A "Rayleigh-like" SRF adjustment would use the fact that most of the TOA signal at 440nm will come from Rayleigh scattering and the Rayleigh scattering efficiency with wavelength is sloped in the opposite direction of the SRF slopes in each detector (higher at shorter wavelength, lower at longer wavelength). Using a unique "Rayleigh-like" correction as a function of wavelength for each sensor could adjust their responses to reflect a more square SRF. However, as of this work, this has not been applied at the L1B stage.

Your recommendation may work well, too. Understanding the impact of this differential SRF on the L1B is still an active area of research in our group. This effect is not seen in HARP CubeSat or HARP2 SRFs.

We decided to remove this discussion for scope. We are not using the 440nm band in the following intercomparison, so speculating on potential corrections here will only distract. We left it as: "Figure 7b also shows a differential SRF for the AirHARP 440nm band, which is likely due to manufacturer error in the thin-film coating for the AirHARP prism interfaces or detector stripe filters. This 440nm SRF differential is unique to AirHARP; we see no evidence of this in the HARP CubeSat or HARP2 440nm designs (Sienkiewicz et al. 2024, *in prep*). We are pursuing several corrections for the AirHARP 440 nm spectral differential at the detector level and L1B stage, though further details are beyond the scope of this work."

Line 461: Leading and trailing edges suggest something to do with flight direction. Blue edge or short-wavelength edge, and red edge or long-wavelength edge would be better terminology in my opinion.

We apologize for the confusion. Short-wavelength and long-wavelength edge are reasonable substitutions. We will address this in the revision.

We modified the text to say: "subscripts *i* and *f* denoting the shorter- and longer-wavelength edges of the spectral band"

Line 469. I think you mean Eq. (15).

Line 475: Do you mean solar zenith angle rather than view zenith angle?

Yes, both of these will be addressed in the revision.

Both are corrected in-text.

Line 560: Since this refers to testing in late 2022, were the referenced measurements acquired?

This was also brought up by reviewer #2. We have decided to remove this reference and date from the paper. While yes, the data was acquired, we prefer to focus on AirHARP here, and leave HARP2 results on POLBOX to a future publication.

This remains our position. The reference and date were removed from the paper.

Line 611: Change "RSB" to "RSP".

This was corrected.

Lines 686-688: I would have expected the sunglint pattern to have a steeper variation in signal as a function of view angle than the desert scene, so I don't follow the argument that the ocean signal is more robust against angular misregistration.

This comment was about the actual surface features that are present at the Rosamond Dry Lake, versus potentially smoother ocean surface (especially off-glint). The Dry Lake has some spatial variation that could impact the intercomparison in this way. Off-glint ocean, for example, maybe less so. Of course, this depends on ocean scene – heavy phytoplankton loading or high winds that chop up the ocean surface may create similar variations, especially near glint. Thank you for your insight on this - we may remove this statement in the revision. It may require more explanation than what it is worth.

We decided to remove this statement in the revision, but because the analysis contributing to Figure 10 changed substantially.

Figure 4 caption: Perhaps change "only the SNR remains" to "only pixel-to-pixel variations due to noise remains".

This was changed in the caption for Figure 5.

General comment: Directional measurements of radiance, multiplied by pi and normalized by illumination irradiance, are referred to as "reflectance" throughout the paper. Per Schaepman-Strub et al. (2006), Rem. Sens. Environ. 103, 27-42 and Nicodemus et al. (1977), NBS publication, reflectance is a ratio of fluxes (exitance/irradiance), hence a measure of albedo. In the

nomenclature defined in these publications, the quantity referred to as reflectance in McBride et al. is a "reflectance factor" for illumination at normal incidence.

Thank you for this observation. We will make the necessary changes and add these references.

All uses of reflectance factor were made explicitly obvious in-text. However, the analysis leading to Figure 10 uses TOA reflectances (reflectance factor/cos(SZA)) in order to harmonize the RSP error model (defined using TOA reflectance) with the rest of the AirHARP and RSP radiometry measurements.